# PVN-mPFC OT projections modulate pup-directed pup care or attacking in virgin mandarin voles

**Lu Li, Yin Li, Caihong Huang, Wenjuan Hou, Zijian Lv, Lizi Zhang, Yishan Qu, Yahan Sun, Kaizhe Huang, Xiao Han, Zhixiong He\*, Fadao Tai\***

Shaanxi Normal University, Xi'an, China

**Abstract** Many species of animals exhibit caregiving or aggression toward conspecific offspring. The neural mechanisms underlying the infanticide and pup care remain poorly understood. Here, using monogamous mandarin voles (*Microtus mandarinus*), we found that more oxytocin (OT) neurons in the paraventricular nucleus (PVN) were activated during pup caring than infanticide. Optogenetic activation of OT neurons in the PVN facilitated pup caring in male and female mandarin voles. In infanticide voles, optogenetic activation of PVN OT cells or PVN-medial prefrontal cortex (mPFC) OT projection fibers prolonged latency to approach and attack pups, whereas inhibition of these OT neurons or projections facilitated approach and infanticide. Optogenetic activation of PVN OT neuron projections to the mPFC in males shortened the latency to approach and retrieve pups and facilitated the initiation of pup care, but produced no effects on pup-care females. In addition, OT release in the mPFC increased upon approaching and retrieving pups, and decreased upon attacking pups. Intraperitoneal injection of OT promoted pup care and inhibited infanticide behavior. It is suggested that the OT system, especially PVN OT neurons projecting to mPFC, modulates pup-directed behaviors and OT can be used to treat abnormal behavioral responses associated with some psychological diseases such as depression and psychosis.

**\*For correspondence:**
hezhixiong@snnu.edu.cn (ZH);
taifadao@snnu.edu.cn (FT)

**Competing interest:** The authors declare that no competing interests exist.

## eLife assessment

This **important** work provides insights into the neural mechanisms regulating specific parental behaviors. By identifying a key role for oxytocin-synthesizing cells in the paraventricular nucleus of the hypothalamus and their projections to the medial prefrontal cortex in promoting pup care and inhibiting infanticide, this study advances our understanding of the neurobiological basis of these contrasting behaviors in male and female mandarin voles. The evidence supporting the authors' conclusions is **solid**, and this work should be of interest to researchers studying neuropeptide control of social behaviors in the brain.

## Introduction

Both paternal and maternal care are critical to the survival as well as the physical and mental well-being of the offspring (*He et al., 2019*). Although we know a great deal about the neural mechanisms underlying maternal care, the neural substrates of paternal behavior remain elusive because of the lack of an ideal animal model of paternal care. Only in some monogamous rodents (e.g., prairie voles), canids, and primates do males assist and spend a great deal of energy caring for pups (*Malcolm, 2015*; *Mendoza and Mason, 1986*; *Rosenfeld et al., 2013*). However, some male rodents without reproductive experience also show paternal care toward alien pups, while some others ignore or even attack alien pups (*Dai et al., 2022*). These different pup-directed behavioral responses are based on

their physiological and environmental states, and the killing of young conspecifics by sexually inexperienced mammals is a widespread phenomenon among different animal taxa (*Lukas and Huchard, 2014*). Infanticide is thought to benefit the infanticide by promoting their own reproduction (*Hrdy, 1974*). In laboratory mice, male mice without pairing experience typically display infanticide (*Svare and Mann, 1981*), but males are able to shift from infanticide to pup care when they have the opportunity to encounter their own offspring (*Elwood, 1977*). Compared to our extensive understanding of the maternal circuit, little is known about the neural substrates underlying female infanticide. The virgin mandarin voles (*Microtus mandarinus*) naturally exhibit biparental care and infanticide in both females and males that provide an ideal animal model to reveal the mechanism underlying paternal care in males and infanticide in females.

Oxytocin (OT) is well known as a key hormone for initiating and maintaining maternal care (*Yoshihara et al., 2018*), which is primarily synthesized in the paraventricular nucleus (PVN) and supraoptic nucleus (SON), among which the PVN plays an important role in initiating maternal care in rats (*Munetomo et al., 2016*). There is evidence to suggest that OT not only regulates maternal motivation, but also mediates paternal behavior (*Bales et al., 2004*). When mouse fathers were exposed to their pups, OT neurons in the PVN were specifically activated and they also showed more aggression toward intruders to protect their own pups (*Shabalova et al., 2020*). Compared with rodents, similar neuropeptides and hormones are involved in paternal behavior in non-human primates (*Woller et al., 2012*). Similar to other mammals, paternal care only exists in a few primate species (*Dulac et al., 2014*; *Woller et al., 2012*). A study on marmoset monkeys showed that fathers have higher levels of OT secretion in the hypothalamus than non-fathers (*Woller et al., 2012*), and intraventricular infusion of OT reduces the tendency of marmoset fathers to refuse to transfer food to their young offspring (*Saito and Nakamura, 2011*). However, where and how pathways of OT neurons regulate pup care and infanticide behavior remain largely unknown.

The medial prefrontal cortex (mPFC) is involved in attention switching, decision-making, behavioral flexibility, and planning, making it potentially crucial for rapidly expressing pup care or infanticide behavior. A study on virgin male and female mice found that the mPFC lesion (targeting the prelimbic cortex) significantly affected a number of females and males showing pup cares and infanticide. Here, 50%females in the lesioned group exhibited maternal while 100% of sham-operated groups show maternal care, whereas the 100% lesioned males exhibited infanticide, 83% of control males showed infanticide (*Alsina-Llanes and Olazábal, 2021*). It has been reported that the mPFC is highly activated in human mothers when they hear cries from their babies (*Lorberbaum et al., 2002*). The mPFC is also activated when rat mother contacts their pups first time (*Fleming and Korsmit, 1996*), while the damage of the mPFC disrupts pup-retrieving and -grooming behavior in rats (*Afonso et al., 2007*). In rodents and humans, the mPFC was activated during the process of caring offspring (*Hernández-González et al., 2005*; *Seifritz et al., 2003*). A study in rat mothers indicated that inactivation or inhibition of neurons in the mPFC largely reduced pup retrieval and grouping (*Febo et al., 2010*). A subsequent study on firing patterns in the mPFC of rat mother suggested that sensory-motor processing carried out in the mPFC may affect decision-making of maternal care to their pups (*Febo, 2012*). Examining different regions of the mPFC (anterior cingulate [Cg1], prelimbic [PrL], infralimbic [IL]) of new mother identified a role for the IL cortex in biased preference decision-making in favor of the offspring (*Pereira and Morrell, 2020*). A study on rats suggests that the IL and Cg1 subregion in mPFC are the motivating circuits for pup-specific biases in the early postpartum period (*Pereira and Morrell, 2011*), while the PrL subregion is recruited and contributes to the expression of maternal behaviors in the late postpartum period (*Pereira and Morrell, 2011*). In addition, a large number of neurons in the mPFC express oxytocin receptors (OTRs) (*Smeltzer et al., 2006*). OT in the circulation of mice can act on the mPFC to increase social interaction and maternal behavior (*Munesue et al., 2021*). Although there are some studies on the role of the mPFC in pup care, the involvement of mPFC OT projections in pup cares and infanticide requires further research. Thus, we hypothesized that PVN OT neuron projections to mPFC may causally control paternal or infanticide behaviors.

Based on the potential antagonistic effects between pup care and infanticide behavior in neural mechanisms (*Dulac et al., 2014*; *Kohl et al., 2017*; *Mei et al., 2023*) as well as the potential role of PVN-to-mPFC OT projections in pup care and infanticide, a combination of methods, including immunohistochemistry, optogenetics, fiber photometry, and intraperitoneal injection of OT, were used to reveal neural mechanisms underlying paternal care and infanticide. We found that PVN OT neurons

regulated the expression of pup care and infanticide, and further identified the involvement of PVN-to-mPFC OT projections in paternal care and infanticide. Collectively, these findings establish a regulatory role for PVN-to-mPFC OT neurons in the expression of pup-directed behaviors and suggest potential targets for the future development of intervention strategies against psychiatric disorders associated with infanticide such as depression and psychosis.

## Results

### Pup-care behavior activates more OT⁺ cells than infanticide in PVN

In order to observe the activated OT neurons in virgin voles during pup care and infanticide behaviors, we co-stained OT and c-Fos on brain slices from voles exhibiting different behaviors using the immunofluorescence method (*Figure 1a*). Histological analysis showed no difference in the number of OT or c-Fos-positive cells between the pup care and infanticide groups of female (*Figure 1b and c*, *Figure 1—source data 1*) and male (*Figure 1e and f*, *Figure 1—source data 1*) voles. Approximately 11% (male) and 14% (female) of OT cells expressed c-Fos during pup caring, whereas only about 3% (males) and 7% (females) of OT neurons were labeled by c-Fos during infanticide (female: $t(6) = 5.173$, $p<0.01$, $d = 3.658$, *Figure 1d*, *Figure 1—source data 1*; male: $t(6) = 2.607$, $p<0.05$, $d = 1.907$, *Figure 1g*, *Figure 1—source data 1*). In male and female voles, more OT neurons were activated during pup caring than infanticide (*Figure 1d and g*). In addition, females displaying pup care and infanticide showed higher merge rates of OT and c-Fos than males displaying the same behaviors ($F(1,12) = 5.002$, $p=0.045$, $\eta^2 = 0.294$, *Figure 1d and g*, *Figure 1—source data 1*).

### Effects of optogenetic activation of PVN OT neurons on pup-directed behaviors

To reveal the causal role of PVN OT neuron in the regulation of pup care and infanticide behaviors, the effects of optogenetic activation of PVN OT neurons on pup-directed behaviors were investigated (*Figure 2a–c*). Over 89% of CHR2 expression overlapped with OT neurons, indicating high specificity of the CHR2 virus (*Figure 2d*, *Figure 2—source data 1*). 473 nm light stimulation increased c-Fos expression in the CHR2 virus-infected brain region that validated the effect of optogenetic activation (*Figure 3—figure supplement 1a-c*). We found that optogenetic activation of PVN OT cells significantly reduced latency to approach (CHR2: OFF vs. ON, $F(1, 7) = 11.374$, $p<0.05$, OFF/ON: $\eta^2 = 0.592$, *Figure 2o*, *Figure 2—source data 1*) and retrieve pups (CHR2: OFF vs. ON, $F(1, 4) = 14.755$, $p<0.05$, OFF/ON: $\eta^2 = 0.156$, *Figure 2p*, *Figure 2—source data 1*) and prolonged crouching time (CHR2: OFF vs. ON, $F(1, 7) = 60.585$, $p<0.001$, OFF/ON: $\eta^2 = 0.419$, *Figure 2s*, *Figure 2—source data 1*) in pup-care males, but had no effect on females (*Figure 2j–n*, *Figure 2—source data 1*), nor in the control virus group. Optogenetic activation of these neurons significantly reduced the latency to approach and attack pups in male (approach: CHR2: OFF vs. ON, $F(1, 5) = 185.509$, $p<0.0001$, OFF/ON: $\eta^2 = 0.552$, *Figure 2h*, *Figure 2—source data 1*; infanticide: CHR2: OFF vs. ON, $F(1, 5) = 59.877$, $p<0.01$, OFF/ON: $\eta^2 = 0.526$, *Figure 2i*, *Figure 2—source data 1*) and female voles (approach: CHR2: OFF vs. ON, $F(1, 6) = 64.810$, $p<0.001$, OFF/ON: $\eta^2 = 0.915$, *Figure 2f*, *Figure 2—source data 1*; infanticide: CHR2: OFF vs. ON, $F(1, 6) = 75.729$, $p<0.001$, OFF/ON: $\eta^2 = 0.940$, *Figure 2g*, *Figure 2—source data 1*) displaying infanticide behaviors, whereas they had no effect on the control virus group (*Figure 2f–l*, *Figure 2—source data 1*). Further, we conducted a two-way rmANOVA on the CHR2 data set for both sexes and found that pup-care females exhibited shorter latencies to approach (OFF: gender simple effect, $F(1,13) = 5.735$, $p=0.032$, $\eta^2 = 0.306$, *Figure 2j and o*) and retrieve pups (OFF: gender simple effect, $F(1,10) = 13.040$, $p=0.005$, $\eta^2 = 0.566$, *Figure 2k and p*) than males (*Figure 2—source data 1*). These results suggest that activation of PVN OT neurons facilitates pup-care behavior and significantly inhibits infanticide behavior.

### Effects of optogenetic inhibition of PVN OT neurons on pup-directed behaviors

To further verify the roles of PVN OT neurons on pup-induced behavior, we optogenetically inhibited OT cells by eNpHR virus and tested pup-directed behaviors (*Figure 3a–c*). More than 90% of neurons expressing eNpHR overlapped with OT-positive neurons, indicating high specificity of eNpHR virus infection (*Figure 3d*, *Figure 3—source data 1*). 589 nm light stimulation to eNpHR virus-infected brain

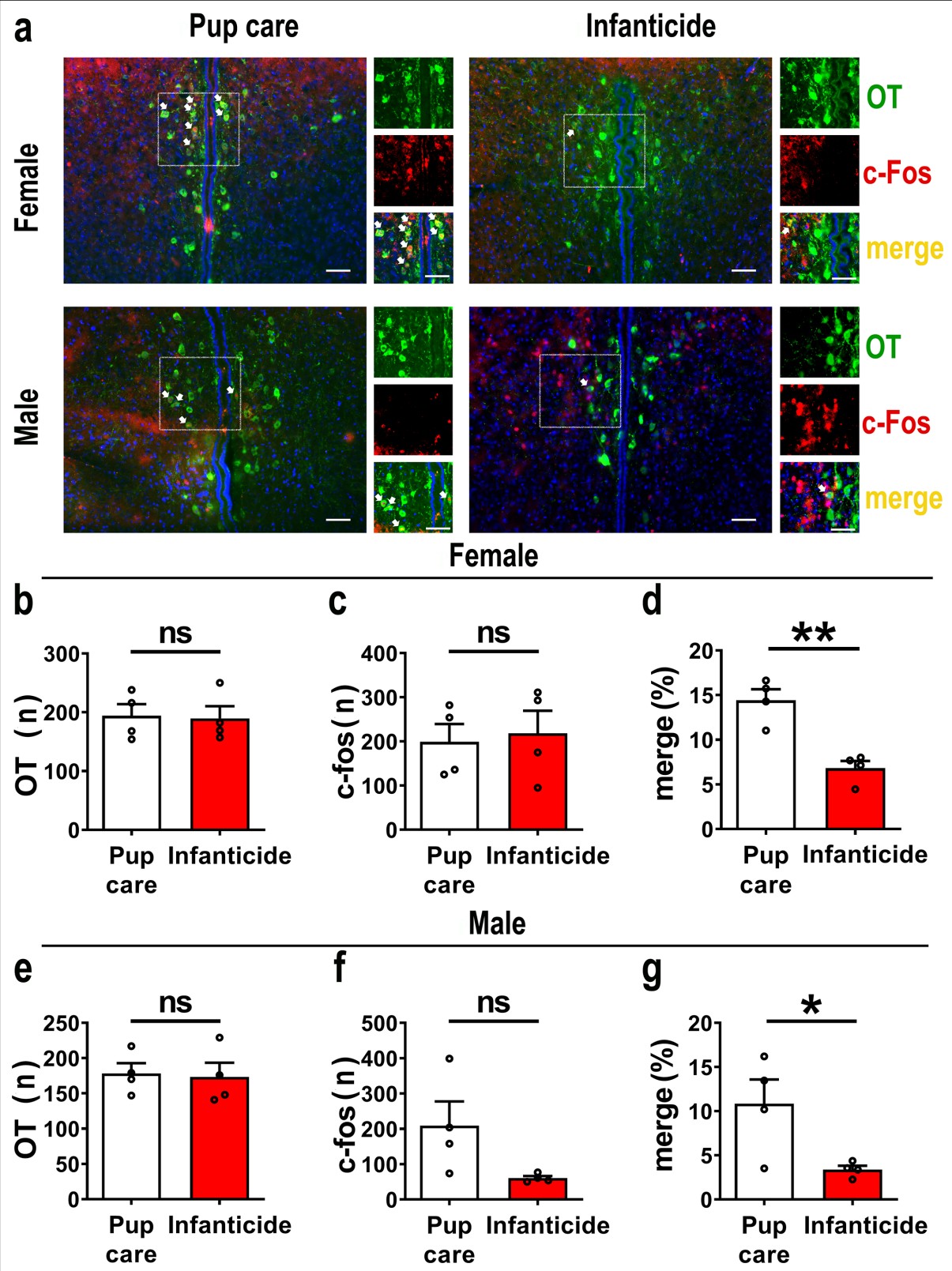

**Figure 1.** Activated oxytocin (OT) neurons in the paraventricular nucleus (PVN) of mandarin voles during pup care (n = 4) and infanticide (n = 4).
(**a**) Representative histological images of OT (green) and c-Fos (red)-positive cells in PVN: Blue, DAPI; yellow, merged cells. Enlarged views of the boxed area are shown on the right of each image, and white arrows indicate the overlap of OT and c-Fos-positive cells. Objective: ×20. Scale bars, 50 μm. (**b**) Number of OT-positive cells in pup care and infanticide female voles. (**c**) Number of c-Fos-positive cells in pup care and infanticide female voles. (**d**)

*Figure 1 continued on next page*

Figure 1 continued

Percentage of c-Fos-expressing cells in OT cells of PVN from pup care and infanticide female voles. **p<0.01. Independent-samples *t*-tests. (**e**) Number of OT cells in pup care and infanticide male voles. (**f**) Number of c-Fos cells in pup care and infanticide male voles. (**g**) Percentage of c-Fos-expressing cells in OT cells of PVN from pup care and infanticide male voles. *p<0.05. Independent-samples *t*-tests. Error bars indicate SEM.

The online version of this article includes the following source data for figure 1:

**Source data 1.** Statistical results of the number of OT-positive cells, the number of c-Fos-positive cells, and the percentage of OT and c-Fos merged neurons in the total PVN OT neurons in female and male pup care and infanticide voles.

regions reduced c-Fos expression verifying the effectiveness of opotogenetic inhibition via eNpHR virus (*Figure 3—figure supplement 1d–f*). Inhibition of PVN OT neurons showed no significant effect on pup-care behavior in male and female voles that spontaneously exhibited pup-caregiving behaviors (*Figure 3j–s*, *Figure 3—source data 1*). For both male and female voles in the infanticide group, optogenetic inhibition significantly shortened the latency to approach (female: $F_{(1, 5)} = 1331.434$, p<0.0001, OFF/ON: $\eta^2 = 0.980$, *Figure 3f*, *Figure 3—source data 1*; male: $F_{(1, 5)} = 10.472$, p<0.05, OFF/ON: $\eta^2 = 0.690$, *Figure 3h*, *Figure 3—source data 1*) and attack pups (female: F 1, 5) = 291.606, p<0.0001, OFF/ON: $\eta^2 = 0.991$, *Figure 3g*, *Figure 3—source data 1*; male: $F_{(1, 5)} = 46.901$, p<0.01, OFF/ON: $\eta^2 = 0.837$, *Figure 3i*, *Figure 3—source data 1*. In addition, we performed a two-way rmANOVA on the eNpHR group data for both sexes and found that pup-care females exhibited shorter latency to approach (gender main effect $F_{(1,10)} = 62.131$, p<0.0001, $\eta^2 = 0.861$, *Figure 3j and o*) and retrieve (gender main effect $F_{(1,10)} = 137.393$, p<0.0001, $\eta^2 = 0.932$, *Figure 3k and p*) than males (*Figure 3—source data 1*). These results suggest that inhibition of OT neurons in the PVN significantly facilitates infanticide behavior.

## Changes in OT release upon pup-directed behaviors

The results of the optogenetic manipulation demonstrated that PVN OT neurons regulated pup-induced behavior. We next detected the OT release in the mPFC during pup-induced behavior by OT1.0 sensor (*Figure 4a–c*). Pup-caring female and male voles showed a significant increase (female: $F_{(1.958, 13.708)} = 45.042$, p<0.001, $\eta^2 = 0.865$; male: $F_{(5, 35)} = 24.057$, p<0.01, $\eta^2 = 0.775$) in the signal for OT1.0 sensors upon approaching (female: p<0.01; male: p<0.05) and retrieving (female: p<0.01; male: p<0.05), whereas there was no significant difference in the signal at the onset of other behaviors compared with the signal before the introduction of the pups (*Figure 4f–m*, *Figure 4—source data 1*). In addition, we compared the signals of OT1.0 sensors in the pup-caring voles upon the first, second, and third approaches to the pups (female: $F_{(2, 14)} = 10.917$, p<0.01, $\eta^2 = 0.609$; male: $F_{(2, 14)} = 13.351$, p<0.01, $\eta^2 = 0.656$), OT release peaked at the first approach and tended to decrease thereafter (*Figure 4d and e*, *Figure 4—source data 1*). In infanticide female and male voles, OT release decreased upon attacking in infanticide males ($F_{(1.117, 7.822)} = 85.803$, p<0.001, $\eta^2 = 0.838$) and females ($F_{(1.068, 7.479)} = 36.336$, p<0.001, $\eta^2 = 0.925$) (*Figure 4n–r*, *Figure 4—source data 1*). In addition, no significant changes in signals were detected from the individuals with control AAV2/9-hSyn-OTmut during pup-directed behaviors (*Figure 4—figure supplement 1*). In addition, no changes in OT release were detected while subjects were exposed to objects with a similar size and shape as the pup (*Figure 4—figure supplement 2*). Besides, we found that pup-care females showed higher AUC per second than males during approaching (gender simple effect $F_{(1,14)} = 27.740$, p=0.000119, $\eta^2 = 0.665$, *Figure 4l and m*, *Figure 4—source data 1*) and retrieving (gender simple effect $F_{(1,14)} = 11.695$, p=0.004, $\eta^2 = 0.455$, *Figure 4l and m*, *Figure 4—source data 1*) the pups. These results indicate that mPFC OT release significantly increased upon approaching and retrieving in pup-care voles, but decreased upon attacking pups in infanticide voles.

## Effects of optogenetic activation of PVN OT neuron fibers in the mPFC on pup-directed behaviors

Previous experiments found that OT release in the mPFC changed upon pup-directed behavior. To manipulate the neural circuit, we first verified oxytocin projections from PVN to the mPFC. We injected retrogradely labeled virus in the mPFC and observed the overlap of virus with OT in the PVN (*Figure 5*), and we also counted the PVN OT neurons projecting to mPFC and found that approximately 45.16 and 40.79% of cells projecting from PVN to the mPFC were OT-positive, and approximately 18.48 and

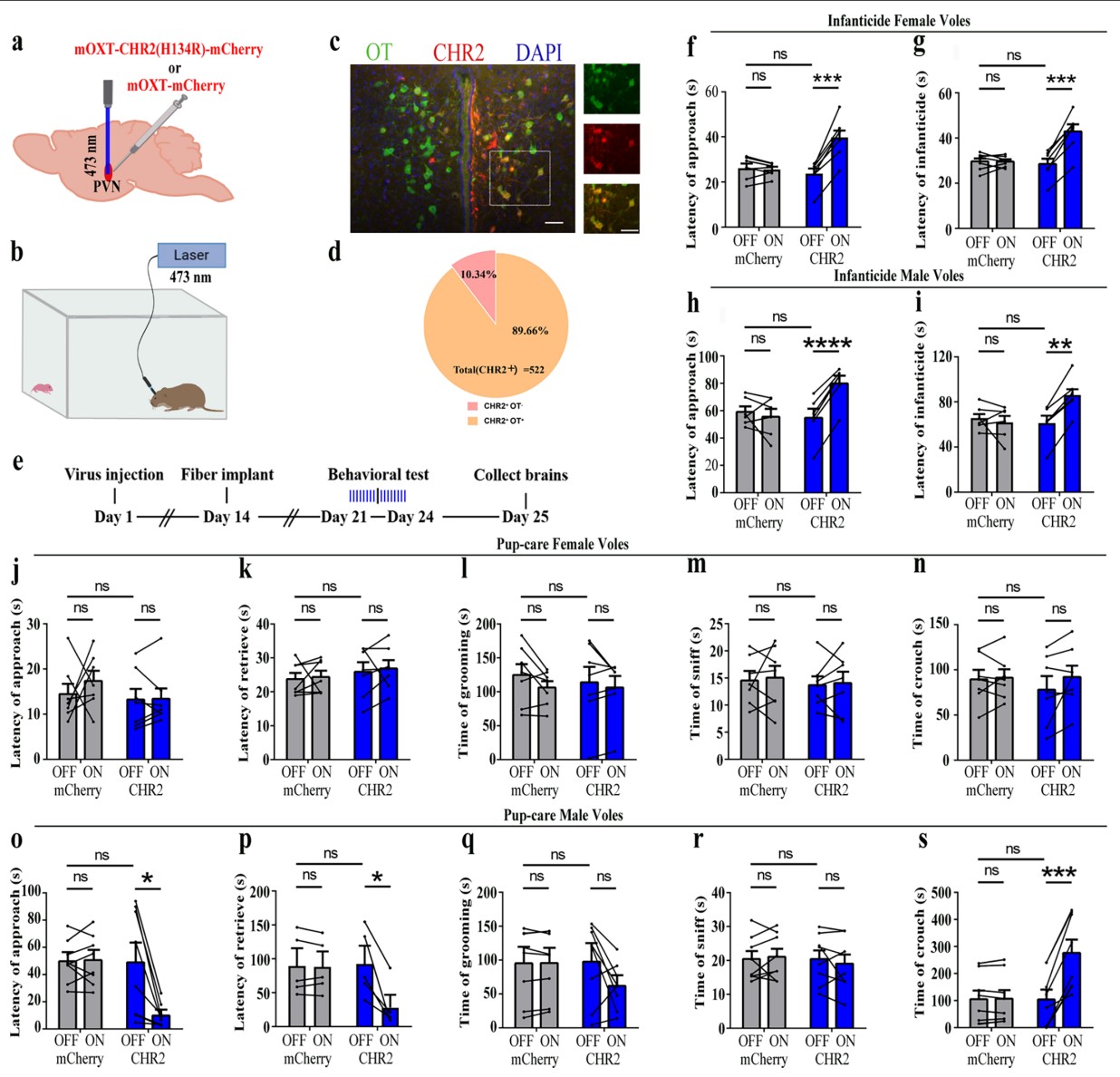

**Figure 2.** Effects of optogenetic activation of paraventricular nucleus (PVN) oxytocin (OT) neurons on pup-directed behaviors. (**a**) Schematic of virus injection and optical fiber implantation. (**b**) Schematic diagram of the behavioral test. (**c**) Representative histological images of CHR2 (red) expression and OT staining (green), enlarged view of the boxed area is on the right side. Blue, DAPI. Objective: ×20. Scale bars, 50 µm. (**d**) Statistics of the specificity of CHR2 expression in three voles; more than 89% of CHR2-positive neurons overlapped with OT-positive neurons. (**e**) Timeline of the experiment. (**f–i**) Approach (**f, h**) and infanticide (**g, i**) latency of seven female (n = 7) and six male (n = 6) voles in mCherry (control virus group) and CHR2 groups. **p<0.01 vs. CHR2 OFF; ***p<0.001 vs. CHR2 OFF; ****p<0.0001 vs. CHR2 OFF. Two-way rmANOVA (factors: treatment × stimulus). (**j–n**) Latency to approach (**j**), latency to retrieve (**k**), grooming time (**l**), sniffing time (**m**), and crouching time (**n**) of seven pup-care female voles (n = 7) in female control virus and CHR2 groups. (**o–s**) Latency to approach (**o**), latency to retrieve (**p**), grooming time (**q**), sniffing time (**r**), and crouching time (**s**) of eight pup-care male voles (n = 8) in male control virus and CHR2 groups. *p<0.05 vs. CHR2 OFF, ***p<0.001 vs. CHR2 OFF. Two-way rmANOVA (factors: treatment × stimulus). Error bars indicate SEM.

The online version of this article includes the following source data for figure 2:

**Source data 1.** Statistical results of the number of cells expressing only CHR2 and co-expressing CHR2 and OT in the PVN of three voles with injection of optogenetic virus, the latency to approach and attack pups in infanticide voles, and the latency to approach, retrieve, duration of grooming, sniffing, and crouching in pup-care voles.

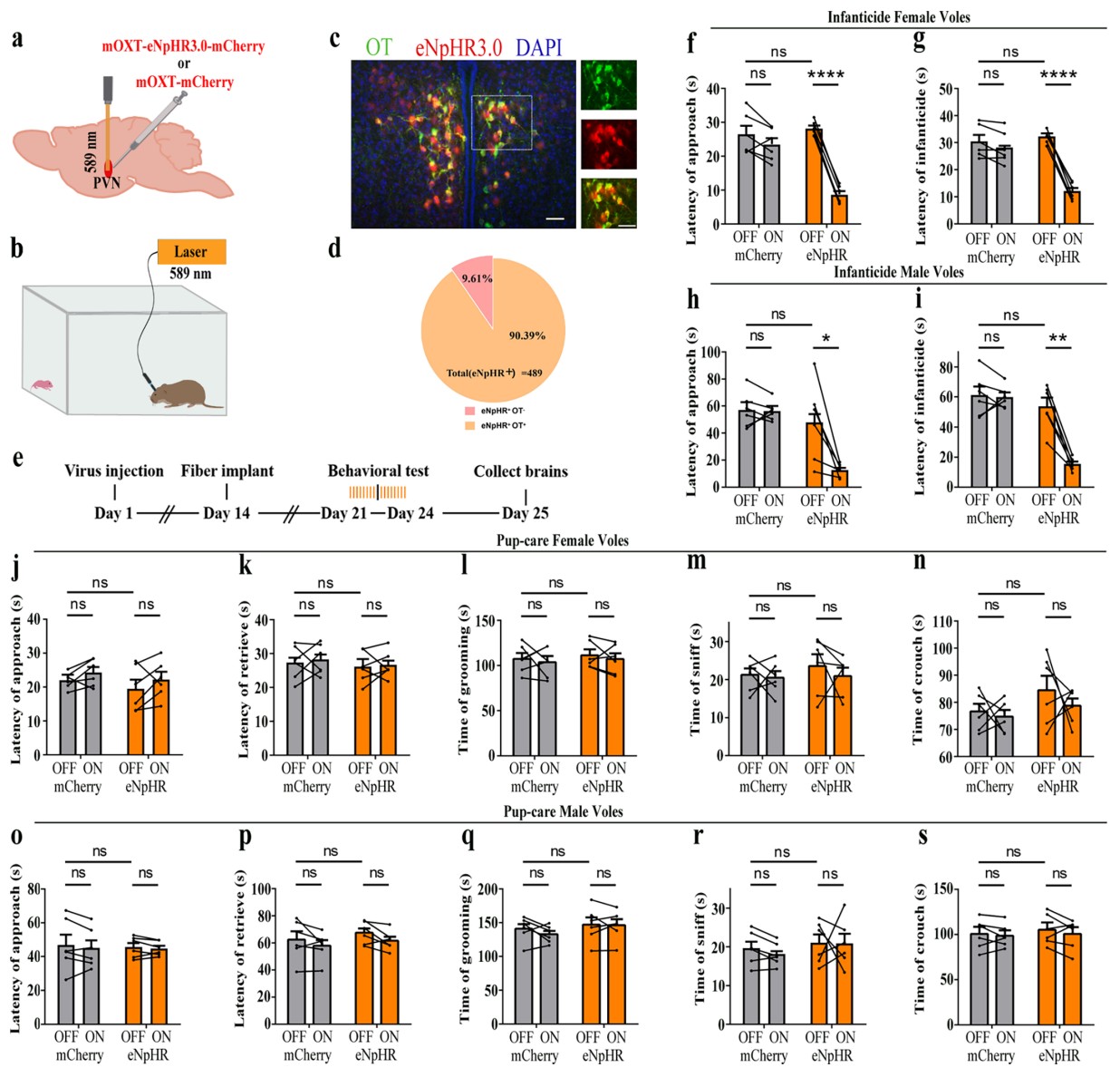

**Figure 3.** Effects of optogenetic inhibition of paraventricular nucleus (PVN) oxytocin (OT) neurons on pup-directed behaviors. (**a**) Schematic of virus injection and optical fiber implantation. (**b**) Schematic diagram of the behavioral test. (**c**) Representative histological images of OT staining (green) and eNpHR (red) expression, enlarged view of the boxed area is shown on the right side. Blue, DAPI. Objective: ×20. Scale bars, 50 µm. (**d**) Statistics of the specificity of CHR2 expression in three voles; more than 90% of eNpHR expression overlapped with OT. (**e**) Timeline of the experiment. (**f–i**) Approach (**f, h**) and infanticide (**g, i**) latency of six female (n = 6) (**f, g**) and six male (n = 6) (**h, i**) infanticide voles. *p<0.05 vs. eNpHR OFF; **p<0.01 vs. eNpHR OFF; ****p<0.0001 vs. eNpHR OFF. Two-way rmANOVA (factors: treatment × stimulus). (**j–n**) Approach latency (**j**), retrieval latency (**k**), grooming time (**l**), sniffing time (**m**), and crouching time (**n**) in six control mCherry (n = 6) and eNpHR (n = 6) groups of pup-care female voles. (**o–s**) Approach latency (**o**), retrieval latency (**p**), grooming time (**q**), sniffing time (**r**), and crouching time (**s**) in six control mCherry (n = 6) and eNpHR (n = 6) groups of pup-care male voles. Error bars indicate SEM.

The online version of this article includes the following source data and figure supplement(s) for figure 3:

**Source data 1.** Statistical results of the number of cells expressing only eNpHR and co-expressing eNpHR and OT in the PVN of three voles with injection of optogenetic virus, the latency to approach and attack pups in infanticide voles, and the latency to approach, retrieve pups, duration of grooming, sniffing, and crouching in pup-care voles.

**Figure supplement 1.** Light-induced c-Fos expression overlapping with mCherry, CHR2, or eNpHR.

**Figure supplement 1—source data 1.** Statistical results of light-induced c-Fos expression overlapping with mCherry, CHR2, or eNpHR.

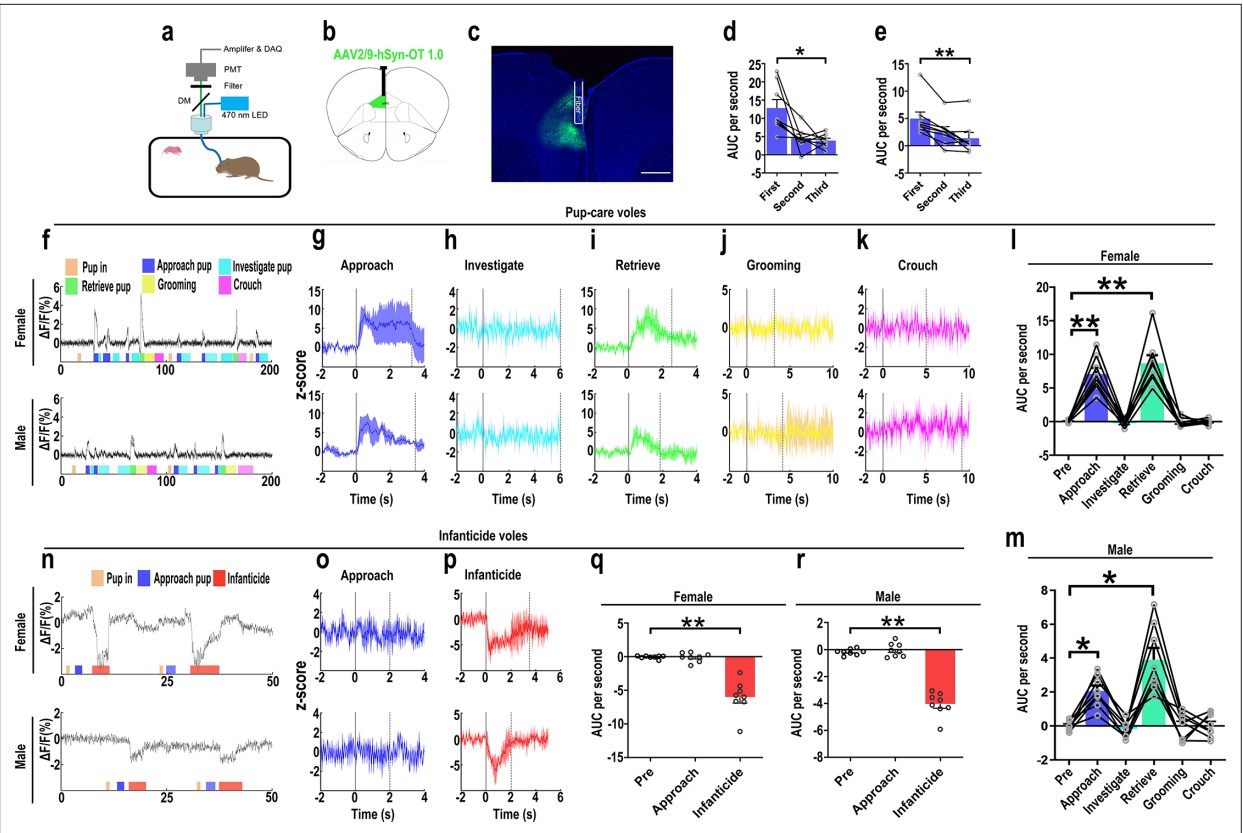

**Figure 4.** Oxytocin (OT) release in the medial prefrontal cortex (mPFC) upon pup-directed behaviors. (**a**) Recording instrument settings. (**b**) Illustrations of viral expression and optical fiber location. (**c**) Representative histological image of OT1.0 sensor (green) and optical fiber locations. Blue, DAPI. Objective: ×4. Scale bars, 500 μm. (**d, e**) Area under the curve (AUC) per second for pup-care female (**d**) and male (**e**) voles approaching pups for the first, second, and third time (n = 8). *p<0.05 vs. first. **p<0.01 vs. first. One-way rmANOVA. (**f**) Representative ΔF/F traces in pup-care female (**f**, top) and male (**f**, bottom) voles during interaction with pups. (**g–k**) Post-event histograms (PETHs) of z-score of OT1.0 sensor for the following pup-directed behaviors: approach (**g**), investigate (**h**), retrieve (**i**), grooming (**j**), and crouch (**k**). (**l, m**) The mean AUC of z-scores for pup-care female (**l**) and male (**m**) voles across various pup-directed behaviors (n = 8). Female: **p<0.01 vs. approach. p<0.01 vs. retrieve. Male: *p<0.05 vs. approach. p<0.05 vs. retrieve. One-way rmANOVA. (**n**) Representative ΔF/F traces in infanticide female (**n**, top) and male (**n**, bottom) voles during interaction with pups. (**o, p**) PETHs of z-score of OT1.0 sensor for approach and infanticide in infanticide voles. (**q, r**) The mean AUC of z-score of OT1.0 sensor for pre-pup exposure, approach, and infanticide in infanticide female (**q**) and male (**r**) voles (n = 8). **p<0.01 vs. infanticide. One-way rmANOVA. Error bars indicate SEM.

The online version of this article includes the following source data and figure supplement(s) for figure 4:

**Source data 1.** Area under the curve per second for pre-pup exposure, approach, and infanticide behaviors in infanticide voles and area under the curve per second for first, second, and third approaches to pups, as well as pre-pup exposure, approach, investigation, retrieval, grooming, and crouching behaviors in pup-care voles.

**Figure supplement 1.** Recordings of OTmut sensor signals in the medial prefrontal cortex (mPFC) on oxytocin (OT) release.

**Figure supplement 1—source data 1.** Area under the curve per second for pre-pup exposure, approach, and infanticide behaviors in infanticide voles and area under the curve per second for pre-pup exposure, approach, investigation, retrieval, grooming, and crouching behaviors in pup-care voles.

**Figure supplement 2.** Recordings of OT1.0 sensor signals for investigating object.

**Figure supplement 2—source data 1.** Area under the curve per second for pre-object exposure, approach, and investigating in female and male voles.

18.89% of OT cells in the PVN projected to the mPFC in females and males, respectively (*Figure 5—figure supplement 1*). Then, we tested whether optogenetic activation of the PVN OT neuron projection fibers in the mPFC affects pup-induced behaviors (*Figure 6a–d*). Similar to previous results, in the pup-caring group, activation of the fibers facilitated approaching (F(1,5) = 23.915, p<0.01, OFF/ON: $\eta^2$ = 0.760) and retrieving (F(1,5) = 39.664, p<0.01, OFF/ON: $\eta^2$ = 0.907) in male voles (*Figure 6o–s, Figure 6—source data 1*), but had no effect on females (*Figure 6j–n, Figure 6—source data 1*). In

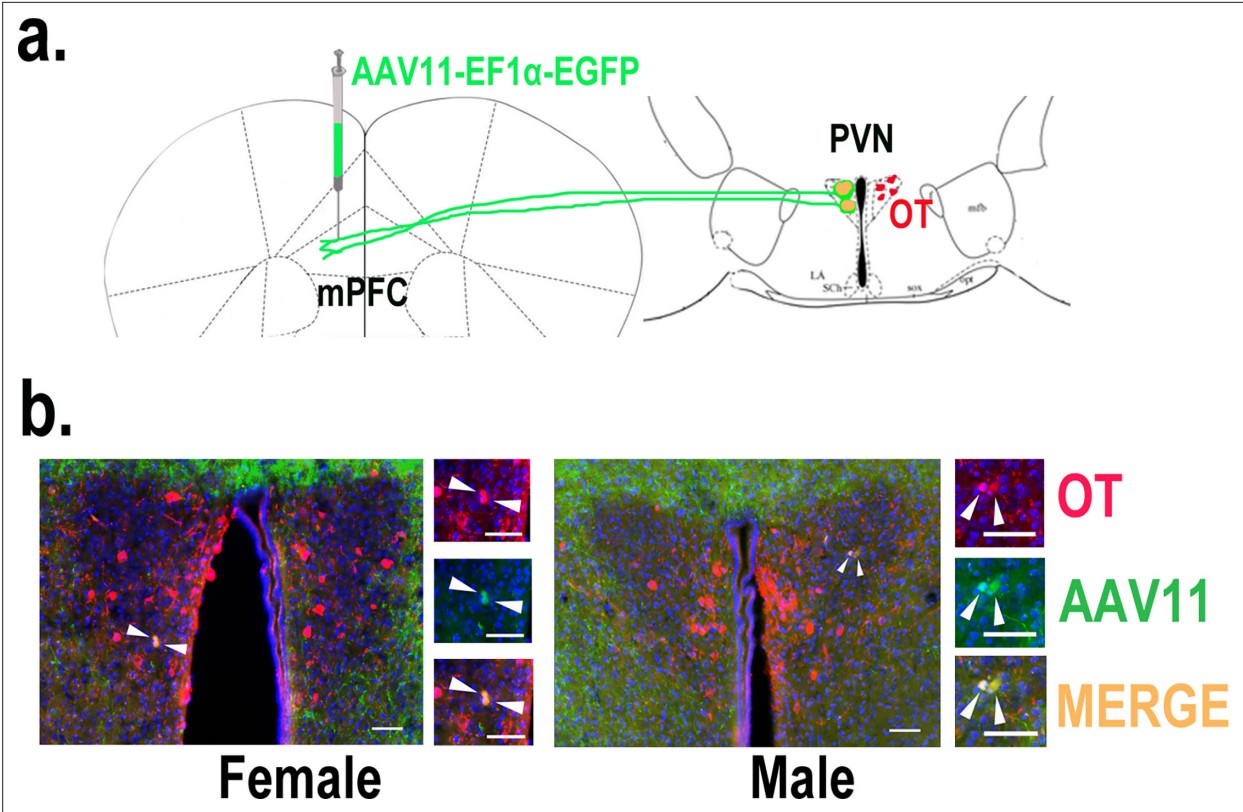

**Figure 5.** Determination of paraventricular nucleus (PVN) to medial prefrontal cortex (mPFC) oxytocin projection. (**a**) Schematic diagram of mPFC virus injection and oxytocin (OT) staining. (**b**) Histological pictures of OT (red) and AAV11 (green) co-staining in male and female. Yellow, merged; blue, DAPI. Objective: ×20. Scale bars, 50 μm.

The online version of this article includes the following source data and figure supplement(s) for figure 5:

**Figure supplement 1.** Number of paraventricular nucleus (PVN) oxytocin (OT) neurons projecting to medial prefrontal cortex (mPFC).

**Figure supplement 1—source data 1.** Counts of AAV11, OT, and co-expressed cells.

male and female infanticide group voles, optogenetic activation of the PVN OT neuron projection fibers prolonged latency to approach (female: F(1,5) = 37.094, p<0.01, OFF/ON: $\eta^2$ = 0.875; male: F(1,5) = 74.718, p<0.001, OFF/ON: $\eta^2$ = 0.889) and attack pups (female: F(1,5) = 38.347, p<0.01, OFF/ON: $\eta^2$ = 0.877; male: F(1,5) = 61.589, p<0.01, OFF/ON: $\eta^2$ = 0.910) (*Figure 6f–i*, *Figure 6— source data 1*). These results suggest that activation of the PVN OT neurons to mPFC projection promoted the onset of pup-care behavior in pup-care male voles and inhibited the onset of infanticide behavior in infanticide voles.

## Optogenetic inhibition of the PVN OT neuron projection fibers promoted infanticide

We then optogenetically suppressed the projection fibers from PVN OT neurons to mPFC and observed changes in pup-directed behaviors (*Figure 7a–d*). Similar to the results of the PVN OT neurons inhibition, we found that optogenetic inhibition of the PVN OT neuron projection fibers promoted approach (F(1,5) = 119.093, p<0.001, OFF/ON: $\eta^2$ = 0.877) and infanticide (F(1,5) = 112.501, p<0.001, OFF/ON: $\eta^2$ = 0.885) in infanticide male (*Figure 7h and i*, *Figure 7—source data 1*) and female voles (approach: F(1,5) = 280.031, p<0.0001, OFF/ON: $\eta^2$ = 0.853; infanticide: F(1,5) = 268.694, p<0.0001, OFF/ON: $\eta^2$ = 0.838) (*Figure 7f and g*, *Figure 7—source data 1*). For pup-care male and female voles, inhibition of these fibers did not significantly affect their pup-care behaviors (*Figure 7j–s*, *Figure 7—source data 1*). To validate the effectivity of fiber optogenetic inhibition, we combined optogenetic inhibition with OT1.0 sensors and recorded a decrease in OT release upon

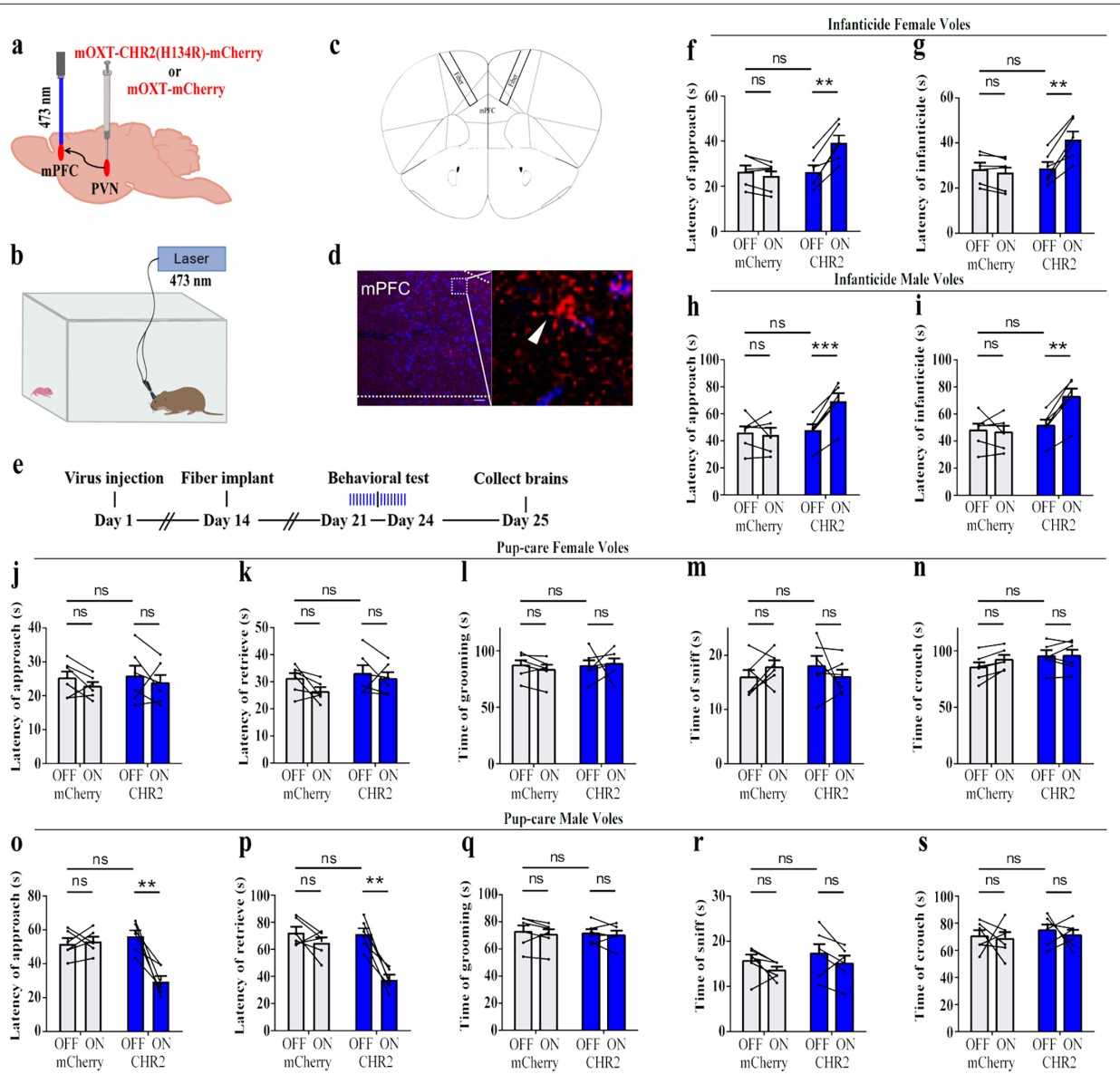

**Figure 6.** Effects of optogenetic activation of the paraventricular nucleus (PVN) oxytocin (OT) neuron projection fibers on pup-directed behaviors. (a) Schematic of virus injection and optical fiber implantation. (b) Schematic diagram of the behavioral test. (c) Illustration of optical fiber implantation in the target brain region. (d) Representative histological pictures of the fiber position and projection fibers. Blue, DAPI. Objective: ×20. Scale bars, 50 μm. (e) Timeline of the experiment. (f–i) Changes in approach latency (f, h) and infanticide latency (g, i) of female (f, g) and male (h, i) infanticide voles in CHR2 and control mCherry group before and after delivery of light (n = 6). **p<0.01 vs. CHR2 OFF; ***p<0.001 vs. CHR2 OFF. Two-way ANOVA (factors: treatment × stimulus). (j–n) Approach latency (j), retrieval latency (k), grooming time (l), sniffing time (m), and crouching time (n) in control mCherry and CHR2 group of pup-care female voles (n = 6). (o–s) Approach latency (o), retrieval latency (p), grooming time (q), sniffing time (r), and crouching time (s) in control mCherry and CHR2 group of pup-care male voles (n = 6). **p<0.01 vs. CHR2 OFF. Two-way rmANOVA (factors: treatment × stimulus). Error bars indicate SEM.

The online version of this article includes the following source data for figure 6:

**Source data 1.** Statistical results of the latency to approach and attack pups in infanticide voles, and the latency to approach, retrieve pups, duration of grooming, sniffing, and crouching in pup-care voles.

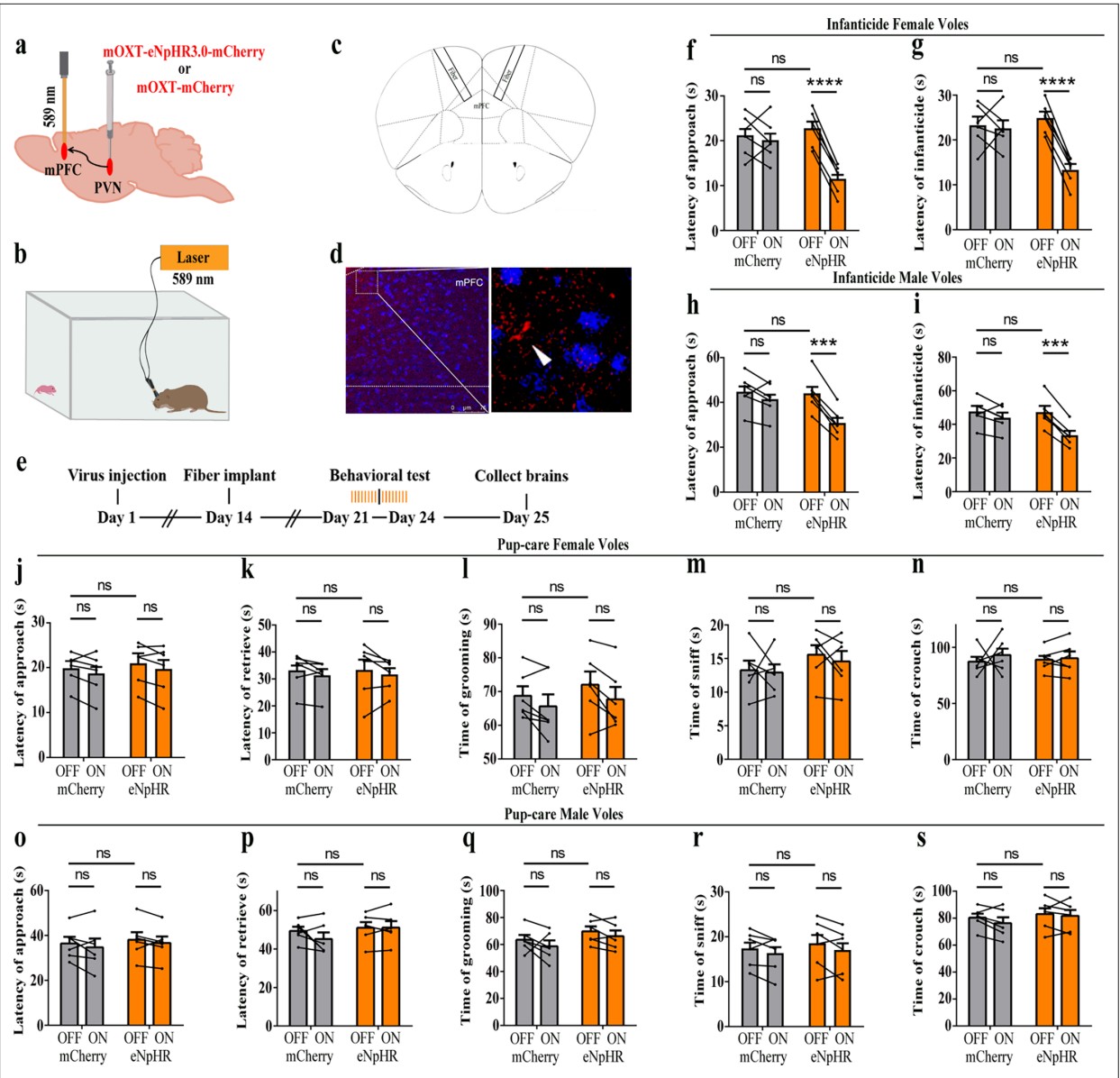

**Figure 7.** Optogenetic inhibition of the paraventricular nucleus (PVN) oxytocin (OT) neuron projection fibers promoted the onset of infanticide. (a) Illustration of virus injection and optical fiber implantation. (b) Schematic of the behavioral test. (c) Diagram of optical fiber implantation in the target brain region. (d) Representative histological pictures of the fiber location and projection fibers. Blue, DAPI. Objective: ×20. Scale bars, 75 μm. (e) Timeline of the experiment. (f–i) Changes in approach (f, h) and infanticide (g, i) latency of female and male infanticide voles in eNpHR (n = 6) and control mCherry groups (n = 6) before and after light delivery. ***p<0.001 vs. eNpHR OFF. ****p<0.0001 vs. eNpHR OFF. Two-way rmANOVA (factors: treatment × stimulus). (j–n) Approach latency (j), retrieval latency (k), grooming time (l), sniffing time (m), and crouching time (n) in control mCherry (n = 6) and eNpHR (n = 6) group of pup-care female voles. (o–s) Approach latency (o), retrieval latency (p), grooming time (q), sniffing time (r), and crouching time (s) in control mCherry (n = 6) and eNpHR group (n = 6) of pup-care male voles. Error bars indicate SEM.

The online version of this article includes the following source data and figure supplement(s) for figure 7:

**Source data 1.** Statistical results of the latency to approach and attack pups in infanticide voles, and the latency to approach and retrieve pups, duration of grooming, sniffing, and crouching in pup-care voles.

**Figure supplement 1.** Recordings of oxytocin (OT) release during optogenetic inhibition of OT fibers in the medial prefrontal cortex (mPFC).

**Figure supplement 1—source data 1.** Statistics of AUC before and after optogenetic inhibition.

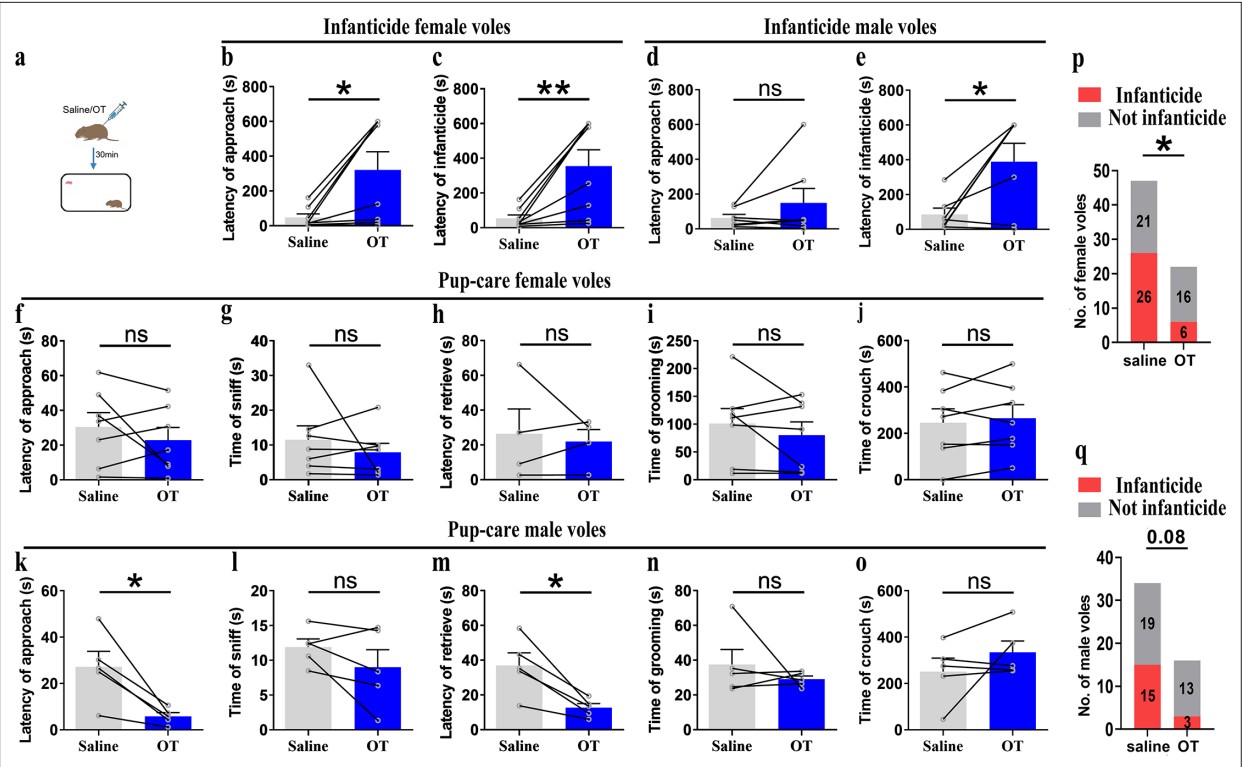

**Figure 8.** Pup-directed behaviors before and after intraperitoneal delivery of oxytocin (OT). (**a**) Diagram of intraperitoneal OT delivery. (**b, c**) Approach (**b**) and infanticide latency (**c**) of infanticide female voles (n = 8). *p<0.05. Paired-samples *t*-test. **p<0.01. Wilcoxon signed-rank test. (**d, e**) Approach (**d**) and infanticide latency (**e**) of infanticide male voles (n = 7). *p<0.05. Paired-samples *t*-test. (**f–j**) Approach latency (**f**, n = 7), sniffing time (**g**, n = 7), latency to retrieve (**h**, n = 4), grooming time (**i**, n = 7), and crouching time (**j**, n = 7) before and after delivery of OT in pup-care female voles. (**k–o**) Approach latency (**k**), sniffing time (**l**), latency to retrieve (**m**), grooming time (**n**), and crouching time (**o**) before and after delivery of OT in pup-care male voles (n = 5). *p<0.05. Paired-samples *t*-test. (**p, q**) Changes in infanticide rates in female (**p**) and male (**q**) voles after administration of saline and OT. *p<0.05. Pearson chi-square test. Error bars indicate SEM.

The online version of this article includes the following source data for figure 8:

**Source data 1.** Statistical results of the latency to approach and attack pups in infanticide voles, and the latency to approach and retrieve pups, duration of grooming, sniffing, and crouching in pup-care voles.

inhibition of fibers (***Figure 7—figure supplement 1***). These results suggest that optogenetic inhibition of the PVN OT neuron projection fibers promotes the onset of infanticide behavior in infanticide voles.

## Intraperitoneal injection of OT

For pre-clinic purposes, we tested the effect of peripheral administration of OT on pup-directed behavior (***Figure 8a***). Delivery of OT promoted approach (t(4) = 3.737, p<0.05, d = 1.335) and retrieval (t(4) = 4.190, p<0.05, d = 2.04) in pup-care male voles (***Figure 8k and m***), while it had no significant effect on the pup-directed behaviors in females (***Figure 8f–j***). For infanticide voles, there was no significant prolongation of approach latency (***Figure 8d***), but a significant extension of infanticide latency of males after delivery of OT (t(6) = –2.988, p<0.05, d = 1.345, ***Figure 8e***). In infanticide female voles, both the latency to approach (Z = –2.380, p<0.05, d = 5.891, ***Figure 8b***) and attack pups (t (7) = –3.626, p<0.01, d = 1.063, ***Figure 8c***) were significantly prolonged by the delivery of OT. In addition, we integrated data from our pre-test of the effects of OT on the number of infanticide voles and found that intraperitoneal injection of OT significantly reduced the number of infanticide female ($\chi^2$ = 4.740, p<0.05, odds ratio [OR] = 0.303, ***Figure 8p***) and male voles ($\chi^2$ = 3.039, p=0.081, OR = 0.292, ***Figure 8q***). These results indicate that peripheral delivery of OT can promote the onset of pup-care behavior in pup-care male voles and can significantly suppress the infanticide in both sexes. This provides a basis for the application of OT in clinic and wildlife management.

# Discussion

In this study, we used monogamous, highly social mandarin voles to explore the role of PVN OT neurons and PVN^OT-mPFC projections in the modulation of pup care and infanticide behaviors. More OT neurons in the PVN were activated during pup care than infanticide behaviors. Optogenetic activation of the OT neurons in the PVN or OT neuron fibers in the mPFC promoted pup care and inhibited infanticide behavior, whereas inhibition of these neurons and their fibers in the mPFC promoted infanticide. In addition, intraperitoneal administration of OT promoted approach and retrieval of pups in pup-care male voles, and inhibited infanticide in both male and female voles. The present study revealed that the PVN to mPFC OT neural projections regulate pup care and infanticide behavior in virgin mandarin voles.

Firstly, we found that more OT neurons in the PVN were activated during pup care than infanticide behaviors, which is consistent with the well-established prosocial role of OT and its ability to promote pup-care behavior (*Bosch and Young, 2018*). In a previous study on virgin male prairie voles, OT and Fos-colabeled neurons in PVN increased after exposure to conspecific pups and experiencing paternal care (*Kenkel et al., 2012*). In another study of prairie voles, OT and c-Fos-colabeled neurons in the PVN significantly increased after becoming parents, which may be due to a shift from virgin to parents (*Kelly et al., 2017*). Meanwhile, we found that activation of OT neurons in the PVN facilitated pup-care behaviors such as approach, retrieval, and crouching in pup-care male voles, whereas inhibition of OT had no effect on paternal behavior; activation and inhibition of OT neurons in the PVN had no significant effect on pup-care behaviors in pup-care females; and activation of OT neurons in the PVN inhibited pup killing in infanticide voles, whereas the corresponding inhibition of OT neurons in the PVN facilitated infanticide. This finding is consistent with a previous report that silencing OT neurons delayed the retrieval behavior in virgin mice (*Carcea et al., 2021*). Further study found that simply observing dams retrieve pups through a transparent barrier could increase retrieval behavior and PVN OT neuron activity of virgin females (*Carcea et al., 2021*). If OTR knockout mice are used, no pup retrieval occurs after observation, and these results suggest that activation of PVN OT neurons in virgin mice induced by visual signals promoted pup-care behaviors (*Carcea et al., 2021*). In addition, this study further demonstrated that OT in the PVN facilitated the retrieval behavior by modulating the plasticity of the left auditory cortex and amplifying the response of mice to the pup's call (*Carcea et al., 2021*). In another study, we found that the OT neurons in the PVN projecting to the ventral tegmental area (VTA) as well as to the Nac brain region regulate pup-directed behaviors, which may also be accompanied by dopamine release (*He et al., 2021*). This study also supports the findings from optogenetic activation of OT neurons in the PVN in the present study. The results of the present study are also supported by a recent study that chemogenetic activation of OT neurons in the PVN increases pup care and reduces infanticide (*Inada et al., 2022*).

However, manipulation of OT neurons in the PVN produced no significant effects on pup caring in pup-care females. This may be due to the fact that female voles have inherently higher OT neural activity (*Häussler et al., 1990*) and female mice have more OT neurons and OT axon projections than males (*Häussler et al., 1990*), and that there are also significant differences in OTR expression between the two sexes (*Insel et al., 1991*; *Uhl-Bronner et al., 2005*) that possibly shows ceiling effects of the OT system. In the present study, we found that females have more activated OT neurons (*Figure 1d and g*) and released higher levels of OT into the mPFC (*Figure 4d and e*) than males. This sex difference has been reported in other study that activation of OT neurons in the PVN activated noradrenergic neurons in the locus coeruleus by co-releasing OT and glutamate, increased attention to novel objects in male rats, and that this neurotransmission was greater in males than in females (*Wang et al., 2021*). In a study on virgin female mice, pup exposure was found to activate oxytocin and OTR-expressing neurons (*Okabe et al., 2017*). Virgin female mice repeatedly exposed to pups showed shorter retrieval latencies and greater c-Fos expression in the preoptic area (POA), concentrations of OT in the POA were also significantly increased, and the facilitation of alloparental behavior by repeated exposure to pups occurred through the organization of the OT system (*Okabe et al., 2017*). In the present study, we also observed that optogenetic activation of OT neurons in PVN increased crouching behavior in the pup-care male voles, but did not affect grooming time possibly via increase of OT release. This result is in line with a previous study that injection of an OTR antagonist into the medial preoptic area (MPOA) of male voles significantly reduced the total duration of pup-care behavior and increased the latency to approach pups and initiate paternal behavior in male voles

(*Yuan et al., 2019*). This finding suggests that the results of peripherally administered OT and optogenetically activated PVN OT neurons in the present study may also have the involvement of OT-OTR interactions in MPOA. Parturition in experimental animals is accompanied by a decrease in infanticide and the emergence of pup-care behavior, and along with this process OTR expression increased not only on mammary contractile cells, but also in various regions of the brain, such as MPOA, VTA, and olfactory bulb (OB), which are all considered to be important brain regions related to the onset and maintenance of pup-care behaviors (*Bosch and Neumann, 2012*; *Shahrokh et al., 2010*; *Yu et al., 1996*). For example, in a OB study, intra-OB injection of OT antagonist significantly delayed the onset of pup-care behaviors such as retrieving pups, crouching, and nesting in female rats, whereas intra-OB injection of OT in virgin females induced 50% of females to show intact pup-care behaviors (*Yu et al., 1996*). Therefore, the effects of activation of PVN OT neurons may be a result of actions on multiple brain regions involved in the expression of pup-care behavior. Which brain regions that PVN OT neurons project to are involved in pups caring or infanticide needs further studies. Although we used a virus strategy to specifically activate or inhibit PVN OT neurons, other neurochemicals may also be released during optogenetic manipulations because OT neurons may also release other neurochemicals. In one of our previous studies, activation of the OT neuron projections from the PVN, the VTA, as well as to the Nac brain also altered pup-directed behaviors, which may also be accompanied by dopamine release (*He et al., 2021*). In addition, back-propagation of action potentials during optogenetic manipulations may also cause the same behavioral effect as direct stimulation of PVN OT cells. These indirect effects on pup-directed behaviors should also be investigated further in future studies.

Optogenetic activation of the OT neural projection fibers from the PVN to the mPFC facilitated the onset of pup-care behaviors, such as approach and retrieval, in pup-care male voles, whereas inhibition of this circuit had no effect; neither activation nor inhibition had a significant effect in pup-care females; and activation of this neural circuit inhibited infanticide in infanticide voles, whereas the corresponding inhibition of this circuit facilitated infanticide. In addition, we have demonstrated an increase in OT release in pup-care voles upon approaching and retrieving pups, and a decrease in OT release in infanticide voles when infanticide occurs, as recorded by OT sensors in the mPFC. There is some evidence indicating that the mPFC may be an important brain region for OT to exert its effects. In addition to expressing OTRs in the mPFC (*Liu et al., 2005*; *Smeltzer et al., 2006*), the mPFC contains OT-sensitive neurons (*Ninan, 2011*) and receives projections of OT neurons from the hypothalamus (*Knobloch et al., 2012*; *Sofroniew, 1983*). It has been further shown that blocking OTR in the mPFC of postnatal rats by an OTR antagonist delayed the retrieval of pups and reduced the number of pups retrieved by rats, impaired the care of pups, decreased the latency to attack intruders, increased the number of attacks, and increased anxiety in postnatal rats but had no effect on the level of anxiety in virgin rats (*Sabihi et al., 2014*). These results further suggest that OT in the mPFC is involved in the regulation of pup-care behavior and support the results of this study in pup-care voles and infanticide voles from an OTR perspective. Previous studies have shown that optical activation of the mPFC maintains aggression within an appropriate range, with activation of this brain region suppressing aggression between male mice and inhibition of the mPFC resulted in quantitative and qualitative escalation of aggression (*Takahashi et al., 2014*), which is similar to our findings with infanticide voles. For pup-care behavior, it has been reported that the mPFC brain region may be involved in the rapid initiation of pup-care behavior in mice without pairing experience (*Alsina-Llanes and Olazábal, 2020*), which supported the experimental results of pup-care male voles in the present study, but the present study further suggested that OT neurons projecting to mPFC regulated pup-caring and infanticide behavior possibly via increase of OT release in the mPFC. We must also mention the function of the mPFC subregion. In the present study, viruses were injected into the PrL. The PrL and IL regions of the mPFC play different roles in different social interaction contexts (*Bravo-Rivera et al., 2014*; *Moscarello and LeDoux, 2013*). A study has shown that the PrL region of the mPFC contributes to active avoidance in situations where conflict needs to be mitigated, but also contributes to the retention of conflict responses for reward (*Capuzzo and Floresco, 2020*). This may reveal that the suppression of infanticide by PVN to mPFC OT projections is a behavioral consequence of active conflict avoidance. In a study of pain in rats, OT projections from the PVN to the PrL were found to increase the responsiveness of cell populations in the PrL, suggesting that OT may act by altering the local excitation-inhibition (E/I) balance in the PrL (*Liu et al., 2023*). A study of anxiety-related behaviors in male rats suggests that the anxiolytic effects of OT in the mPFC are PL-specific and that

this is achieved primarily through the engagement of GABAergic neurons, which ultimately modulate downstream anxiety-related brain regions, including the amygdala (*Sabihi et al., 2017*). This may provide possible downstream pathways for further research.

Another interesting finding is that peripheral OT administration promoted pup-care behavior in male voles, such as approaching and retrieving pups, but had no significant effect on pup-care behaviors in female voles. This result was supported by a previous study that OT administered peripherally inhibited infanticide in pregnant and reproductively inexperienced females and promoted pup caring in pregnant females (*McCarthy et al., 1986*). Further research has demonstrated that OT in the central nervous system also inhibited infanticide in female house mice (*McCarthy, 1990*). In addition, peripheral administration of OT inhibits infanticide in male mice without pairing experience (*Nakahara et al., 2020*). Moreover, the experience of pairing facilitated the retrieval of pups by increasing peripheral OT levels in both male and female mice (*Lopatina et al., 2011*). This is similar to the findings of pup-care males in the present study, whereas the absence of similar results in females may be due to the differences in OT levels between the sexes (*Tamborski et al., 2016*). Pup-care female voles in this study already showed short latency to approach and retrieve pups before peripheral administration of OT. We also found that peripheral OT administration suppressed infanticide behavior in both male and female voles. It is consistent with a previous study that infanticide in female house mice with no pairing experience and pregnant females with preexisting infanticide was effectively suppressed by subcutaneous injection of OT, while injection of OT also helped pregnant females show care for strange pups (*McCarthy et al., 1986*). Unpaired male mice's aggression toward pups was accompanied by changes in the activity of the vomeronasal neurons, and as males were paired with females and lived together, the activity of these neurons decreased, accompanied by a shift from infanticide to pup-care behavior (*Nakahara et al., 2020*). The OTR expresses throughout the vomeronasal epithelium that allows OT to potentially inhibit infanticide by reducing vomeronasal activity. A previous study found that intraperitoneal injection of OT reduced the activity of the vomeronasal and further validated that OT modulates the activity of vomeronasal neurons by acting on the sensory epithelium to produce behavioral changes by intraperitoneal injection of an OTR antagonist that cannot cross the blood–brain barrier (*Nakahara et al., 2020*). It has recently been shown that peripheral OT was able to cross the blood-brain barrier into the central nervous system to act (*Yamamoto et al., 2019*), meaning that delivery of OT via the periphery may have increased central OT levels and thus exerted an effect. A recent study demonstrated that the secretion of OT in the brain of male mice with no pairing experience facilitated the performance of pup-care behaviors and inhibited infanticide (*Inada et al., 2022*), which also supported the results of the present study. Similar to rodents and non-human primates, there is evidence to suggest that OT contributes to paternal care (*Feldman and Bakermans-Kranenburg, 2017*). Fathers with partners have higher plasma OT levels than non-fathers without partners (*Mascaro et al., 2014*). Intranasal OT treatment increases fathers' play, touch, and social interaction with their children (*Weisman et al., 2012*). Our result provides possible application of OT in the reduction of abnormality in pup-direct behavior associated with psychological diseases in humans such as depression and psychosis (*Milia and Noonan, 2022*; *Naviaux et al., 2020*) and increases of well-being of wildlife.

In summary, these results indicate that the PVN to mPFC OT neural projection is involved in the regulation of pup care and infanticide behavior in virgin mandarin voles. These data provide new insights into the neural circuits underlying OT-mediated pup-directed behaviors.

## Materials and methods

**Key resources table**

| Reagent type (species) or resource | Designation | Source or reference | Identifiers | Additional information |
|---|---|---|---|---|
| Transfected construct (*Mus musculus*) | AAV2/9-mOXT: Promoter-hCHR2(H134R)–mCherry-ER2-WPRE-pA | Shanghai Taitool Bioscience | S0442-9-H20 | Adeno-associated virus construct to transfect and express CHR2 |
| Transfected construct (*M. musculus*) | AAV2/9-mOXT: Promoter-mCherry-pA | Shanghai Taitool Bioscience | S0443-9 | Adeno-associated virus constructs to transfect and express mCherry |

*Continued on next page*

*Continued*

| Reagent type (species) or resource | Designation | Source or reference | Identifiers | Additional information |
|---|---|---|---|---|
| Transfected construct (*M. musculus*) | AAV2/9-mOXT-eNpHR3.0-mCherry-WPRE-hGH-pA | BrainVTA | Cat#PT-2812 | Adeno-associated virus constructs to transfect and express eNpHR3.0 |
| Transfected construct (*M. musculus*) | AAV2/9-hSyn-OT 1.0 | Brain Case | BC-0293 | Adeno-associated virus constructs to transfect and express OT sensor |
| Transfected construct (*M. musculus*) | AAV2/9-hSyn-OTmut | Brain Case | BC-1119 | Adeno-associated virus constructs to transfect and express OTmut sensor |
| Transfected construct (*M. musculus*) | AAV11-EF1α-EGFP | Brain Case | BC-0012 | Adeno-associated virus constructs to transfect and express EGFP |
| Antibody | Mouse anti-OT (monoclonal antibody) | Millipore | Cat# MAB5296; RRID:AB_2157626 | 1:7000 |
| Antibody | Rabbit anti-c-Fos (polyclonal antibody) | Abcam | Cat# ab190289; RRID:AB_2737414 | 1:1000 |
| Antibody | Goat anti-mouse Alexa Fluor 488 | Jackson ImmunoResearch | Cat# 115-545-062; RRID:AB_2338845 | 1:200 |
| Antibody | Goat anti-rabbit Alexa Fluor 488 | Jackson ImmunoResearch | Cat# 111-545-003; RRID:AB_2338046 | 1:200 |
| Antibody | Goat anti-rabbit TRITC | Jackson ImmunoResearch | Cat# 111-025-003; RRID:AB_2337926 | 1:200 |
| Chemical compound, drug | Ready-to-use DAPI | Boster | Cat# AR1177 | |
| Chemical compound, drug | Ready-to-use goat serum | Boster | Cat# AR0009 | |
| Chemical compound, drug | OT | Bachem | Cat# 50-56-6 | |
| Software, algorithm | MATLAB | MathWorks | RRID:SCR_001622 | |
| Software, algorithm | JWatcher | http://www.jwatcher.ucla.edu/ | RRID:SCR_017595 | |
| Software, algorithm | SPSS | IBM | RRID:SCR_002865 | |

## Animals

Mandarin voles were captured from the wild in Henan, China. All laboratory procedures were in accordance with the Guidelines for the Care and Use of Laboratory Animals in China and the regulations of the Animal Care and Use Committee of Shaanxi Normal University. This study protocol was reviewed and approved by the Academic Committee of Shaanxi Normal University, Special Committee of Scientific Ethics, approval no. 2022-041. The virgin mandarin voles (*M. mandarinus*) used in this study were F2 generations that were bred at the Animal Center of Shaanxi Normal University and were kept at 24°C under a 12 hr light-dark cycle (lights on at 8 a.m.) with food and water provided ad libitum. Before the experiments, we exposed the animals to pups, and subjects may exhibit pup care, infanticide, or neglect; we grouped subjects according to their behavioral responses to pups, and individuals who neglect pups were excluded. The stereotactic surgery was performed at the age of 8 weeks of voles. After surgery, the voles were housed with their cage mates. Behavioral tests were carried out 3 weeks after surgery for animal recovery and the viral infection, and 1–5-day-old pups were from other breeders. In case the pups were attacked, we removed them immediately to avoid unnecessary injuries, and the injured pups were euthanized. Sample sizes were determined with reference to previous studies, and all efforts were made to minimize animal suffering.

## Viruses

AAV2/9-mOXT: Promoter-hCHR2(H134R)–mCherry-ER2-WPRE-pA ($8.41 \times 10^{12}$ µg/ml) and AAV2/9-mOXT: Promoter-mCherry-pA ($1.21 \times 10^{13}$ µg/ml) were purchased from Shanghai Taitool Bioscience

Ltd. AAV2/9-mOXT-eNpHR3.0-mCherry-WPRE-hGH-pA ($2.27 \times 10^{12}$ µg/ml) were purchased from BrainVTA (Wuhan, China) LTD. AAV2/9-hSyn-OT 1.0 ($2.06 \times 10^{12}$ µg/ml), AAV2/9-hSyn-OTmut ($2.10 \times 10^{12}$ µg/ml), and AAV11-EF1α-EGFP ($5.00 \times 10^{12}$ µg/ml) were purchased from Brain Case Biotechnology Ltd. For details about the construction of CHR2 and mCherry viruses used in optogenetic manipulation, refer to a previous study in which they constructed an rAAV-expressing Venus from a 2.6 kb region upstream of OT exon 1, which is conserved in mammalian species (*Knobloch et al., 2012*). For details about the construction of the eNpHR 3.0 virus, refer to one study in which the expression of the vector is driven by the mouse OXT promoter, a 1 kb promoter upstream of exon 1 of the OXT gene, which has been shown to induce cell type-specific expression in OXT cells (*Penagarikano et al., 2015*). For details about the construction of OT 1.0 sensor, refer to the research of Professor Li's group (*Qian et al., 2023*). All viruses were dispensed and stored at –80°C.

## Immunohistochemistry

After behavioral tests, serial brain sections were harvested for histological analysis. Anesthetized voles were perfused with 40 ml of PBS and 20 ml of 4% paraformaldehyde. After perfusion, brains were excised and post-fixed by immersion in 4% paraformaldehyde overnight at 4°C. Brains were dehydrated in 20% and then 30% sucrose for 24 hr, respectively, before they were embedded in OCT and cryosectioned into 40 µm slices. Brain slices were rinsed with PBS (10 min) and PBST (PBS and 0.1% Triton X-100, 20 min), blocked in ready-to-use goat serum (Boster, AR0009) for 30 min at room temperature (RT), and then incubated overnight at 4°C with primary antibody in PBST. The primary antibodies used were mouse anti-OT (1:7000, Millipore, MAB5296) and rabbit anti-c-Fos (1:1000, Abcam, ab190289). Following PBST washing (3 × 5 min), sections were incubated with secondary antibodies in PBST for 2 hr (RT) and stained with DAPI, and then washed once more with PBS (3 × 5 min). The secondary antibodies used were goat anti-mouse Alexa Fluor 488 (1:200, Jackson ImmunoResearch, 115-545-062), goat anti-rabbit Alexa Fluor 488 (1:200, Jackson ImmunoResearch, 111-545-003), goat anti-rabbit TRITC (1:200, Jackson ImmunoResearch, 111-025-003), and ready-to-use DAPI staining solution (Boster, AR1177). Finally, the brain slices were sealed with an anti-fluorescent attenuation sealer.

Images were captured with a fluorescent microscope (Nikon) to confirm the viral expression, placements of optic fiber and viruses, and also the number of c-Fos, OT, and virus-positive cells. To analyze the activity of OT neurons (co-expression of c-Fos and OT) among different behaviors and the specificity of viral expression (co-expression of viruses and OT) in the PVN brain region, brain slices of 40 µm were collected consecutively on four slides, each slide had six brain slices spaced 160 µm apart from each other, and counting was performed on one of the slides. Positive cells in the PVN were manually counted based on the Allen Mouse Brain Atlas and our previous studies.

## Stereotaxic surgery

For optogenetic manipulation experiments, CHR2, eNpHR, and mCherry-expressing control virus were stereoaxically injected into the PVN (AP: –0.4 mm, ML: 0.2 mm, DV: 5.3 mm) bilaterally through a Hamilton needle using nanoinjector (Reward Life Technology, KDS LEGATO 130) at 100 nl/min. For optogenetic manipulation, optic fibers (2.5 mm O.D., Reward Life Technology, China) with an appropriate fiber length (PVN: 6 mm, mPFC: 3 mm) were implanted ~100 µm above the PVN and mPFC (AP: 2.2 mm ML: 0.9, DV: 1.89 with a 20° angle lateral to middle) and secured with dental cement (Changshu ShangChi Dental Materials Co, Ltd, 202005). For fiber optometry experiment, optic fibers were inserted ~100 µm above the mPFC (AP: 2.2 mm ML: 0.3, DV: 1.8) after injecting the OT1.0 sensor viruses. The stereotaxic coordinates were determined from three-dimensional brain atlas (*Chan et al., 2007*) and the adjusted coordinates in our lab. The stereotaxic coordinates were determined by the Allen Mouse Brain Atlas and laboratory-corrected data used for voles. Individuals with appropriate viral expression and optical fiber embedding location were included in the statistical analysis, otherwise excluded. The diffusion of central optogenetic viruses and OT1.0 sensors is shown in *Figure 9*.

## Optogenetic

Test animals were injected with 300 nl AAV2/9-mOXT: Promoter-hCHR2(H134R)–mCherry-ER2-WPRE-pA or rAAV-mOXT-eNpHR3.0-mCherry-WPRE-hGH-pA bilaterally into the PVN at 100 nl/min. Control animals were injected with AAV2/9-mOXT: Promoter-mCherry-pA in the same condition. Two

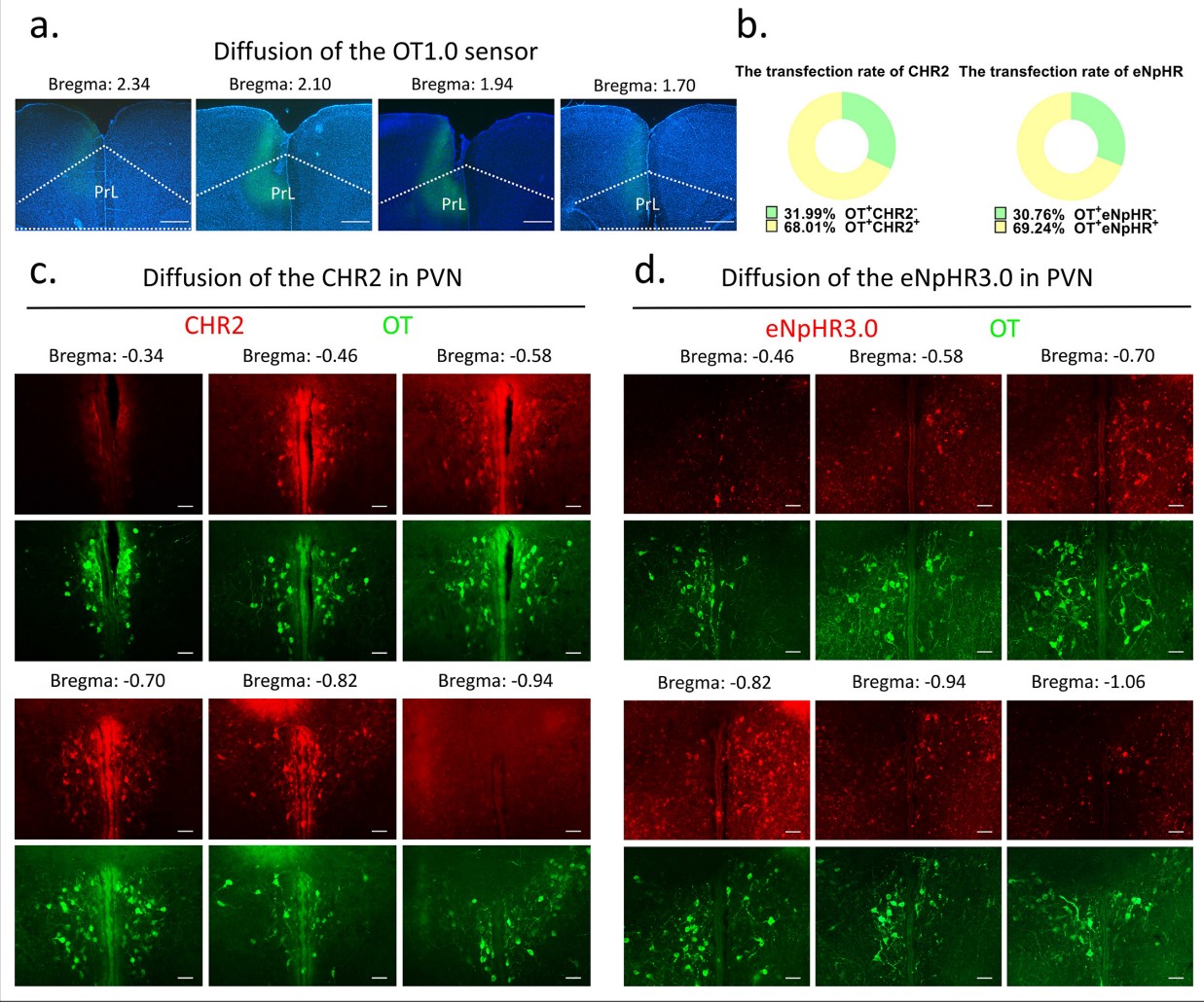

**Figure 9.** Diffusion of centrally injected agents. (**a**) Diffusion of OT1.0 sensor in PrL. Green: OT1.0 sensors; blue: DAPI; objective: ×4. Scale bars: 500 μm. (**b**) The transfection rate of chr2 (n = 3) and eNpHR3.0 (n = 3) in OT cells. (**c**) Diffusion of CHR2 in PVN. Green: OT; red: CHR2; objective: ×20. Scale bars: 50 μm. (**d**) Diffusion of eNpHR3.0 in PVN. Green: OT; red: eNpHR3.0; objective: ×20. Scale bars: 50 μm.

The online version of this article includes the following source data for figure 9:

**Source data 1.** Counting of CHR2 and eNpHR3.0 virus-transfected OT cells.

weeks later, an optic fiber was implanted ~100 μm above the PVN and mPFC bilaterally and was secured using dental cement. After surgery, animals were housed with their cage mates as before. Before the behavioral test, each implanted fiber was connected to a light laser (Newdoon Inc, Aurora 300, 473 nm for activation, 589 nm for inhibition) with a 400 μm patch cord and then we introduced the test animal into the center of the test arena. After the test animal acclimated to the arena for at least 20 min and settled down, we placed a pup in the farthest corner away from the test animal and immediately began recording and applying light stimulation.

In pup-directed pup care behavioral test, light stimulation lasted for 11 min. The parameters used in the optogenetic manipulation of PVN OT neurons were ~3 mW, 20 Hz, 20 ms, 8 s ON, and 2 s OFF, and the parameters used in the optogenetic manipulation of PVN OT neurons projecting to mPFC were ~10 mW, 20 Hz, 20 ms, 8 s ON, and 2 s OFF to cover the entire interaction. In the infanticide behavioral test, the stimulation lasted until the pup was removed. Each vole was tested twice successively, more than 30 min apart, once with the stimulation OFF, once ON. The effect of optogenetic manipulation-induced locomotion on behavioral responses to pups was excluded by recording the total distance traveled by voles without and with light stimulation for 5 min, respectively (***Figure 10***).

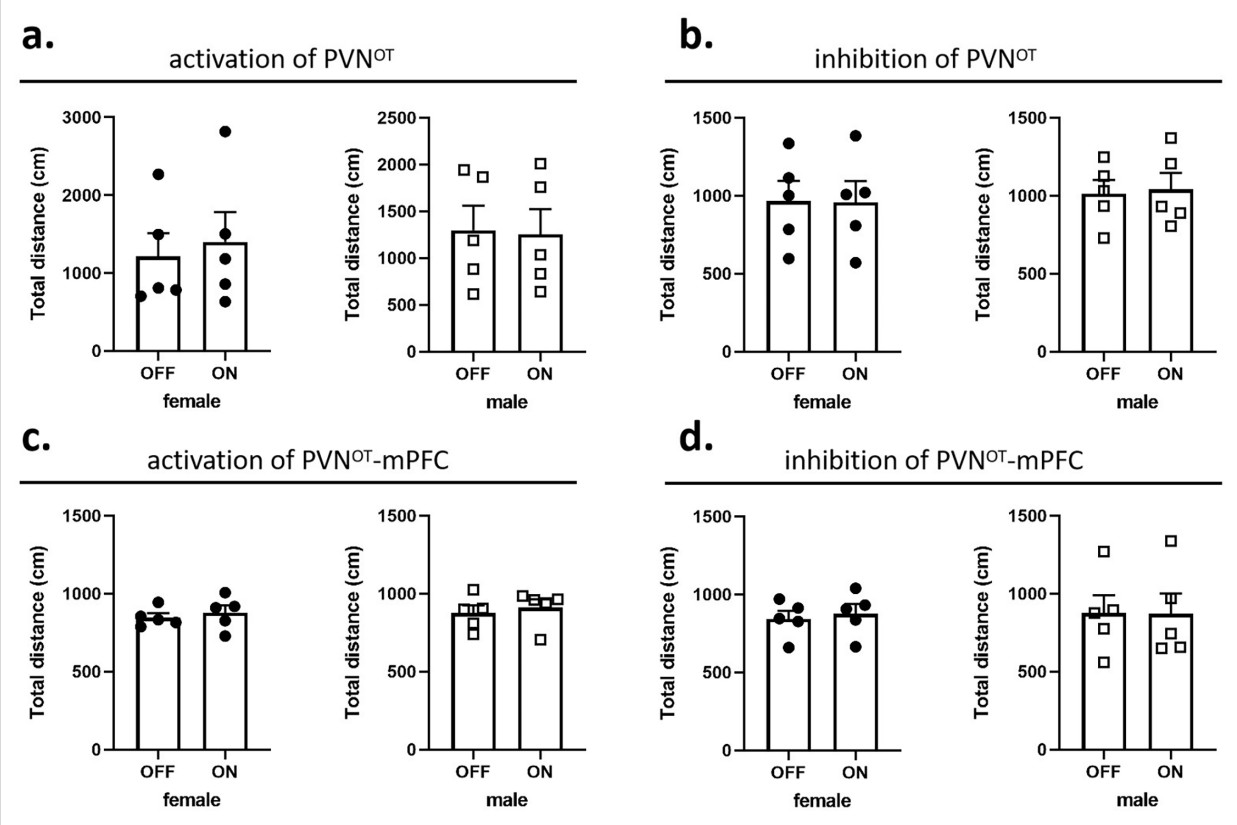

**Figure 10.** Effects of optogenetic manipulations on locomotion of the subjects. (**a**, **b**) Total distance traveled by the subject before and after activation or inhibition of the paraventricular nucleus (PVN) oxytocin (OT) neurons (n = 5). (**c, d**) Total distance traveled by the subject before and after activation or inhibition of the PVN OT neurons projections to the medial prefrontal cortex (mPFC) (n = 5). Error bars indicate SEM. Statistical analyses were performed with paired samples t-tests.

The online version of this article includes the following source data for figure 10:

**Source data 1.** Statistics of distance traveled before and after optogenetic manipulation.

To confirm that ChR2 or eNpHR3.0 stimulation indeed induced neural activation or inhibition, we used light to stimulate the brain through an optical fiber when voles were alone in their home cage, and subsequently determined neural activation or inhibition by c-Fos staining one and a half hours after light stimulation.

## Fiber photometry

To record the fluorescence signals of the OT1.0 sensor during various pup-directed behaviors, virgin voles were anesthetized with 1–2% isoflurane and immobilized on a stereotaxic device (RWD, China). Then, 300 nl AAV9-hSyn-OT1.0 or AAV9-hSyn-OTmut virus was injected into the left side of mPFC (AP: 2.2 mm ML: 0.3, DV: 1.8). After the injection, a 200 μm optical fiber was implanted ~100 μm above the injection site and fixed with dental cement. After 2 weeks of recovery, the optic fiber was connected to the fiber photometry system (QAXK-FPS-LED, ThinkerTech, Nanjing, China) through a patch cable. This system can reduce the effects of motion artifacts by simultaneously recording signals stimulated by a 405 nm light source. To avoid bleaching the sensor, the 470 nm laser power at the tip of the fiber was adjusted to 50 μW. Voles were placed in test cages and allowed to move freely for at least 20 min to acclimate to the environment. Then, a pup was placed in the cage at a distance from the testing vole. If the vole ignored the pup completely or attacked the pup, gently remove the pup and introduce another pup ~60 s later to stimulate more interaction. This process was repeated 3–6 times, and then the vole exhibiting pup-care behavior was allowed to freely interact with the last introduced pup until the vole crouched over the pup for more than 10 s, after which the pups were removed for about 60 s and reintroduced. This latter process was repeated 3–4 times. After the pup

test, we subsequently placed an object (a vole-sized plastic toy) into the cage and recorded four voles that investigated the object 3–6 times.

Videos were recorded with screen-recoding software to synchronize the OT1.0 fluorescence signals and pup-directed behaviors. Fluorescence signals were recorded into MATLAB mat files and analyzed with customized MATLAB code. Data were matched to a variety of behaviors toward pups based on individual trials. The change in signal was displayed as z-scored ΔF/F, which was measured by ($V_{signal}$ - mean ($V_{basal}$))/std ($V_{basal}$). $V_{signa}$ and $V_{basal}$ refer to the recorded values at each time point and the recorded values during the baseline period before the stimuli. The area under the curve (AUC) was calculated based on z-scored ΔF/F matching the duration of the behavior, and the AUC per second was used to compare the different fluorescence signals of behaviors and the baseline.

For the combination of optogenetic inhibition and fiber optometry experiment, optogenetic virus and OT1.0 sensor were injected as described above, optic fibers were inserted above the mPFC (OT1.0 fibers: AP: 2.2 mm ML: 0.2, DV: 1.7; optogenetic fibers: AP: 2.2 mm ML: 2.0, DV: 2.4 with a 45° angle lateral to middle). The signals of OT1.0 sensor were recorded while neurons were optogenetically inhibited.

## Behavioral paradigm and analysis

Animal behaviors in optogenetic experiments were recorded by a camera from the side of a transparent cage. 'Approach pup' was defined as the testing vole faced and walked right up to pup, and the latency to approach was the period from the time the pup was placed in the cage until the vole began to approach the pup. 'Investigate pup' was defined as the vole's nose came into close contact with any part of the pup's body. 'Attack pup' was defined as the vole attacked or bit a pup that can be recognized by the wound, and the latency to attack was the period from the time the pup was placed in the cage until the vole launched an attack. 'Retrieve pup' was defined as from the time a vole picked up a pup using its jaws to the time it dropped the pup at or around the nest, and the latency of retrieval was the time between the pup was put in the cage and the time the vole picked up the pup in its jaws. 'Groom pup' was defined as a vole combed the pup's body surface with its muzzle, accompanied by a rhythmic up-and-down bobbing of the vole's head and displacement of the pup. 'Crouch' was defined as the vole squatted quietly over the pup with no apparent movement. Pup-directed behaviors in optogenetic experiments were scored and analyzed using JWatcher (http://www.jwatcher.ucla.edu/) by an individual blind to experiment design.

## OT treatments

Test virgin voles were acclimatized in their cages for 20 min before being injected intraperitoneally with 0.9% NaCl (1 ml/kg) and pups were introduced 30 min later. The behavioral responses were recorded from the side using a video camera. Thirty minutes after the first record, voles were re-injected intraperitoneally OT 1 mg/kg (*Leuner et al., 2012*) (Bachem, 50-56-6), and pups were introduced 30 min later and behavioral responses were recorded. All behaviors were scored and analyzed using JWatcher.

## Statistics

Parametric tests were used to analyze normally distributed data, and nonparametric tests were used for data that is not normally distributed according to Kolmogorov–Smirnov tests. Independent-samples *t*-tests (two-tailed) were performed to assess number of OT, c-Fos, and merge rate of OT and c-Fos during different pup-directed behaviors (Pup-care vs. Infanticide) and the number of c-Fos-IR-positive neurons (mCherry vs. CHR2; mCherry vs. eNpHR 3.0). The behavioral changes following optogenetic activation and inhibition (factors: treatment × stimulus) of PVN and mPFC were analyzed by two-way repeated-measures ANOVA. One-way ANOVA was used to analyze the AUC changes recorded by the OT sensors in different behaviors. Paired-samples *t*-test (two-tailed) and Wilcoxon signed-ranks test were used to analyze changes in pup-directed behaviors before and after intraperitoneal injection of OT. Pearson chi-square test was used to compare the difference in the number of infanticide voles between the saline and OT groups. All data were presented as mean ± s.e.m., and statistical analyses of data were performed using MATLAB and SPSS 22.0 software.

## Acknowledgements

This research was funded by the STI2030-Maior Projects grant number 2022ZD0205101, the National Natural Science Foundation of China grants numbers 32270510 and 31901082, Natural Science Foundation of Shaanxi Province, China grant number 2020JQ-412, China Postdoctoral Science Foundation grant number 2019M653534, and Fundamental Research Funds for Central University grants numbers GK202301012.

## Additional information

### Funding

| Funder | Grant reference number | Author |
|---|---|---|
| National Natural Science Foundation of China | 32270510 | Fadao Tai |
| National Natural Science Foundation of China | 31901082 | Zhixiong He |
| Natural Science Foundation of Shaanxi Province | 2020JQ-412 | Zhixiong He |
| China Postdoctoral Science Foundation | 2019M653534 | Zhixiong He |
| Fundamental Research Funds for the Central Universities | GK202301012 | Zhixiong He |
| Ministry of Science and Technology of the People's Republic of China | 2022ZD0205101 | Fadao Tai |

The funders had no role in study design, data collection and interpretation, or the decision to submit the work for publication.

### Author contributions
Lu Li, Conceptualization, Data curation, Formal analysis, Methodology, Writing – original draft, Writing – review and editing; Yin Li, Data curation, Investigation, Methodology; Caihong Huang, Yahan Sun, Resources, Investigation, Methodology; Wenjuan Hou, Data curation, Formal analysis, Investigation; Zijian Lv, Resources, Data curation, Methodology; Lizi Zhang, Yishan Qu, Resources, Data curation, Investigation; Kaizhe Huang, Xiao Han, Investigation, Methodology; Zhixiong He, Fadao Tai, Conceptualization, Resources, Supervision, Funding acquisition

### Author ORCIDs
Yishan Qu ⓘ https://orcid.org/0000-0002-1849-6361
Fadao Tai ⓘ https://orcid.org/0000-0002-6804-4179

### Ethics
All laboratory procedures were in accordance with the Guidelines for the Care and Use of Laboratory Animals in China and the regulations of the Animal Care and Use Committee of Shaanxi Normal University. This study protocol was reviewed and approved by the Academic Committee of Shaanxi Normal University, Special Committee of Scientific Ethics, Approval No. 2022-041. In case the pups were attacked, we removed them immediately to avoid unnecessary injuries, and injured pups were euthanized.

Reviewer #1 (Public review): https://doi.org/10.7554/eLife.96543.3.sa1
Reviewer #2 (Public review): https://doi.org/10.7554/eLife.96543.3.sa2
Reviewer #3 (Public review): https://doi.org/10.7554/eLife.96543.3.sa3
Author response https://doi.org/10.7554/eLife.96543.3.sa4

## Additional files

### Supplementary files
• MDAR checklist

### Data availability
All data generated or analysed during this study are included in the manuscript and source data files.

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
