## [Editor Report · eLife assessment]

This **important** work provides insights into the neural mechanisms regulating specific parental behaviors. By identifying a key role for oxytocin-synthesizing cells in the paraventricular nucleus of the hypothalamus and their projections to the medial prefrontal cortex in promoting pup care and inhibiting infanticide, this study advances our understanding of the neurobiological basis of these contrasting behaviors in male and female mandarin voles. The evidence supporting the authors' conclusions is **solid**, and this work should be of interest to researchers studying neuropeptide control of social behaviors in the brain.

---

## [Referee Report · Reviewer #1 (Public review)]

Summary:

This important study investigated the role of oxytocin (OT) neurons in the paraventricular nucleus (PVN) and their projections to the medial prefrontal cortex (mPFC) in regulating pup care and infanticide behaviors in mandarin voles. The researchers used techniques like immunofluorescence, optogenetics, OT sensors, and peripheral OT administration. Activating OT neurons in the PVN reduced the time it took pup-caring male voles to approach and retrieve pups, facilitating pup care behavior. However, this activation had no effect on females. Interestingly, this same PVN OT neuron activation also reduced the time for both male and female infanticidal voles to approach and attack pups, suggesting PVN OT neuron activity can promote pup care while inhibiting infanticide behavior. Inhibition of these neurons promoted infanticide. Stimulating PVN->mPFC OT projections facilitated pup care in males and in infanticide prone voles, activation of these terminals prolonged latency to approach and attack. Inhibition of PVN->mPFC OT projections promoted infanticide. Peripheral OT administration increased pup care in males and reduced infanticide in both sexes. However, some results differed in females, suggesting other mechanisms may regulate female pup care.

Strengths:

This multi-faceted approach provides converging evidence and strengthens the conclusions drawn from the study and make them very convincing. Additionally, the study examines both pup care and infanticide behaviors, offering insights into the mechanisms underlying these contrasting behaviors. The inclusion of both male and female voles allows for the exploration of potential sex differences in the regulation of pup-directed behaviors. The peripheral OT administration experiments also provide valuable information for potential clinical applications and wildlife management strategies.

Weaknesses:

While the study presents exciting findings, there are several weaknesses. The sample sizes used in some experiments, such as the Fos study and optogenetic manipulations, appear to be small, which may limit the statistical power and generalizability of the results.

There is potential effect of manipulating OT neurons on the release of other neurotransmitters (or the influence of other neurochemicals or brain regions) on pup-directed behaviors, especially in females, are not fully explored. Additionally, it is unclear whether back-propagation of action potentials during optogenetic manipulations causes the same behavioral effect as direct stimulation of PVN OT cells. However, the authors now discuss these possibilities. It is also uncertain whether more OT neurons were manipulated in females compared to males. All other comments have been addressed by the authors.

---

## [Referee Report · Reviewer #2 (Public review)]

Summary:

This series of experiments studied the involvement of PVN OT neurons and their projection to the mPFC in pup-care and attack behavior in virgin male and female Mandarin voles. Using Fos visualization, optogenetics, fiber photometry and IP injection of OT the results converge on OT regulating caregiving and attacks on pups. Some sex differences were found in the effects of the manipulations.

Strengths:

Major strengths are the modern multi-method approach and including both sexes of Mandarin vole in every experiment.

Weaknesses:

The few weaknesses include (1) Some experiments' groups have small sample sizes (4-5 animals) which may render some results difficult for others to replicate when different extraneous variables are likely to be present, and (2) the authors discuss PVN OT cell stimulation findings seen in other rodents so the work seems less conceptually novel. Overall, the findings add to the knowledge about OT regulation of pup-directed behavior in male and female rodents, especially the PVN-mPFC OT

---

## [Referee Report · Reviewer #3 (Public review)]

Summary:

Here Li et al. examine pup-directed behavior in virgin Mandarin voles. Some males and females tend towards infanticide, others tend towards pup care. c-Fos staining showed more oxytocin cells activated in the paraventricular nucleus (PVN) of the hypothalamus in animals expressing pup care behaviors than in infanticidal animals. Optogenetic stimulation of PVN oxytocin neurons (with an oxytocin-specific virus to express the opsin transgene) increased pup-care, or in infanticidal voles increased latency towards approach and attack. Suppressing activity of PVN oxytocin neurons promoted infanticide. Use of a recent oxytocin GRAB sensor (OT1.0) showed changes in medial prefrontal cortex (mPFC) signals as measured with photometry in both sexes. Activating mPFC oxytocin projections increased latency to approach and attack in infanticidal females and males (similar to the effects of peripheral oxytocin injections), whereas in pup caring animals only males showed a decrease to approach. Inhibiting these projections increased infanticidal behaviors in both females and males, and no effect on pup caretaking.

Strengths:

Adopting these methods for Mandarin voles is an impressive accomplishment, especially the valuable data provided by the oxytocin GRAB sensor. This is a major achievement and helps promote systems neuroscience in voles.

The authors have done a good job responding to the comments on their preprint. I'd ask them to check their z-scored values, as the mean of a z-scored value should be 0 over time. Also I'm not sure I agree that the fiber photometry system "can automatically exclude effects of motion artifacts"; yes that is a function of imaging at a different wavelength but that process is also prone to error and imperfect.

---

## [Author Response]

The following is the authors’ response to the original reviews.

**Public Reviews:**

**Reviewer #1 (Public Review):**
Summary:This important study investigated the role of oxytocin (OT) neurons in the paraventricular nucleus (PVN) and their projections to the medial prefrontal cortex (mPFC) in regulating pup care and infanticide behaviors in mandarin voles. The researchers used techniques like immunofluorescence, optogenetics, OT sensors, and peripheral OT administration. Activating OT neurons in the PVN reduced the time it took pup-caring male voles to approach and retrieve pups, facilitating pup-care behavior. However, this activation had no effect on females. Interestingly, this same PVN OT neuron activation also reduced the time for both male and female infanticidal voles to approach and attack pups, suggesting PVN OT neuron activity can promote pup care while inhibiting infanticide behavior. Inhibition of these neurons promoted infanticide. Stimulating PVN->mPFC OT projections facilitated pup care in males and in infanticide-prone voles, activation of these terminals prolonged latency to approach and attack. Inhibition of PVN->mPFC OT projections promoted infanticide. Peripheral OT administration increased pup care in males and reduced infanticide in both sexes. However, some results differed in females, suggesting other mechanisms may regulate female pup care.Strengths:This multi-faceted approach provides converging evidence, strengthens the conclusions drawn from the study, and makes them very convincing. Additionally, the study examines both pup care and infanticide behaviors, offering insights into the mechanisms underlying these contrasting behaviors. The inclusion of both male and female voles allows for the exploration of potential sex differences in the regulation of pup-directed behaviors. The peripheral OT administration experiments also provide valuable information for potential clinical applications and wildlife management strategies.Weaknesses:While the study presents exciting findings, there are several weaknesses that should be addressed. The sample sizes used in some experiments, such as the Fos study and optogenetic manipulations, appear to be small, which may limit the statistical power and generalizability of the results. Effect sizes are not reported, making it difficult to evaluate the practical significance of the findings. The imaging parameters and analysis details for the Fos study are not clearly described, hindering the interpretation of these results (i.e., was the entire PVN counted?). Also, does the Fos colocalization align with previous studies that look at PVN Fos and maternal/ paternal care? Additionally, the study lacks electrophysiological data to support the optogenetic findings, which could provide insights into the neural mechanisms underlying the observed behaviors.

In some previous studies (He et al., 2019; Mei, Yan, Yin, Sullivan, & Lin, 2023), the sample size in morphological studies is also small and may be representative. We agree with reviewer’s opinion that results from larger sample size may be more statistically powerful and generalizable. We will pay attention to this issue in the future study. As reviewer suggested, we have added effect size both in the source data and in the main text, including d, η2 and odds ratio. We have added the objective magnification used in the figure legend. The imaging parameters and analysis details for the Fos study have also been added in the revised manuscript. Brain slices of 40 µm thick were collected consecutively on 4 slides, each slide had 6 brain slices spaced 160 µm apart from each other. PVN area were determined based on the Allen Mouse Brain Atlas and our previous study, and Fos, OT and merged positive neurons were counted. Our result about Fos and OT colocalization is consistent with previous study. In a previous study on virgin male prairie voles, OT and Fos colabeled neurons in the PVN increased after exposure to conspecific pups and experiencing paternal care (Kenkel et al., 2012). In another study of prairie voles, OT and c-fos colabeled neurons in PVN significantly increased after becoming parents which may be due to a shift from virgin to parents (Kelly, Hiura, Saunders, & Ophir, 2017). To support the optogenetic findings, we used c-Fos expression as a marker of neuron activity and revealed significant increases/decreases of c-Fos positive neurons induced by optogenetic activation/inhibition (Supplementary Data Fig. 1), and additionally we found that optogenetic inhibition of OT neurons reduced levels of OT release using OT1.0 sensors. Based on these two experiments, we verified that optogenetic manipulation in the present study is validate and results of optogenetic experiment are reliable (Supplementary Data Fig. 5).

The study has several limitations that warrant further discussion. Firstly, the potential effects of manipulating OT neurons on the release of other neurotransmitters (or the influence of other neurochemicals or brain regions) on pup-directed behaviors, especially in females, are not fully explored. Additionally, it is unclear whether back-propagation of action potentials during optogenetic manipulations causes the same behavioral effect as direct stimulation of PVN OT cells. Moreover, the authors do not address whether the observed changes in behavior could be explained by overall increases or decreases in locomotor activity.

We agree with reviewer’s suggestion that several limitations should be discussed. Although we used a virus strategy to specifically activate or inhibit PVN OT neurons, other neurochemical may also be released during optogenetic manipulations because OT neurons may also release other neurochemicals. In one of our previous studies, activation of the OT neuron projections from the PVN to the VTA as well as to the Nac brain also altered pup-directed behaviors, which may also be accompanied by dopamine release (He et al., 2021). In addition, backpropagation of action potentials during optogenetic manipulations may also causes the same behavioral effect as direct stimulation of PVN OT cells. These effects on pup-directed behaviors should also be investigated further in the future study. For the optogenetics experiments, we have referred to some of the previous research (Mei et al., 2023; Murugan et al., 2017), and in our study we have also carried out the verification of the reliability of the methods. To exclude effects of locomotor activity on pup directed behaviors, we also investigated effect of optogenetic manipulations on the locomotor activity of experimental animals and found that optogenetic manipulation did not change levels of locomotor activity (Supplementary Data Fig. 6).

The authors do not specify the percentage of PVN->mPFC neurons labeled that were OT-positive, nor do they directly compare the sexes in their behavioral analysis (or if they did, it is not clear statistically). While the authors propose that the sex difference in pup-directed behaviors is due to females having greater OT expression, they do not provide evidence to support this claim from their labeling data. It is also uncertain whether more OT neurons were manipulated in females compared to males. The study could benefit from a more comprehensive discussion of other factors that could influence the neural circuit under investigation, especially in females.

AAV11-Ef1a-EGFP virus can infect fibers and retrogradely reach to cell body, thus this virus can be used to retrogradely trace neurons. We injected this virus (green, AAV11-Ef1a-EGFP) in the mPFC and observed virus infected and OT (red) positive neuron in the PVN (Yellow), and we also counted the OT neurons that project from PVN to mPFC and found that approximately 45.16% and 40.79% of cells projecting from PVN to the mPFC were OT-positive, and approximately 18.48% and 18.89% of OT cells in the PVN projected to the mPFC in females and males, respectively (Supplementary Data Fig. 4). In addition, as reviewers suggested, we compared the numbers of OT neurons, activated OT neurons (OT and Fos double-labeled neurons) and level of OT release between males and females. We found that females have more activated OT neurons (Figure1, d, g) and released higher levels of OT into the mPFC (Figure 4 d, e) than males. This part has been added in the result and discussion. We did not analyze whether more OT neurons were manipulated in females compared to males, which is indeed a limitation of this study that requires our attention.

As the reviewers suggested, we also discussed other factors that could influence the neural circuit under investigation. In addition to OT neurons, OTR neurons may also regulate behavioral responses to pups. In a study of virgin female mice, pup exposure was found to activate oxytocin and oxytocin receptor expressing neurons (Okabe et al., 2017). Other brain regions such as preoptic area (POA) may also be involved in parental behaviors. For example, virgin female mice repeatedly exposed to pups showed shorter retrieval latencies and greater c-Fos expression in the preoptic area (POA), concentrations of OT in the POA were also significantly increased, and the facilitation of alloparental behavior by repeated exposure to pups occurred through the organization of the OT system (Okabe et al., 2017). A recent study suggests that OT of the PVN is involved in the care of pups by male voles (He et al., 2021). This study suggests that PVN to ventral tegumental area (VTA) OT projections as well as VTA to nucleus accumbens (NAc) DA projections are involved in the care of pups by male voles. Inhibition of OT projections from the PVN to the VTA reduces DA release in the NAc during licking and grooming of pups (He et al., 2021). The effects of these factors on pup-directed responses should also be considered in the future study.

**Reviewer #2 (Public Review):**
Summary:This series of experiments studied the involvement of PVN OT neurons and their projection to the mPFC in pup-care and attack behavior in virgin male and female Mandarin voles. Using Fos visualization, optogenetics, fiber photometry, and IP injection of OT the results converge on OT regulating caregiving and attacks on pups. Some sex differences were found in the effects of the manipulations.Strengths:Major strengths are the modern multi-method approaches and involving both sexes of Mandarin vole in every experiment.Weaknesses:Weaknesses include the lack of some specific details in the methods that would help readers interpret the results. These include:(1) No description of diffusion of centrally injected agents.

Thanks for your professional consideration. Individuals with appropriate viral expression and optical fiber implant location were included in the statistical analysis, otherwise excluded. For optogenetic experiments, the virus (AAV2/9-mOXT-hCHR2(H134R)–mCherry-ER2-WPRE-pA or rAAV-mOXT-eNpHR3.0-mCherry-WPRE-hGH-pA) was designed and constructed to only infect OT neurons, which limited the diffusion of the virus. For fiber photometric experiments, the OT1.0 sensor was largely able to restrict expression within the mPFC brain region, and additionally individuals with incorrect optical fiber embedding position were not included in the statistical analysis. The diffusion of central optogenetic viruses and OT1.0 sensors are shown in the supplemental figure (Supplementary Data Fig. 7).

(2) Whether all central targets were consistent across animals included in the data analyses. This includes that is not stated if the medial prelimbic mPFC target was in all optogenetic study animals as shown in Figure 4 and if that is the case, there is no discussion of that subregion's function compared to other mPFC subregions.

As shown in Figure 4 and in the schematic diagram of the optogenetic experiment, the central targets of virus infection and fiber location remain consistent in the data analysis, otherwise the data would be excluded. In the present study, viruses were injected into the prelimbic (PrL). The PrL and infralimbic (IL) regions of the mPFC play different roles in different social interaction contexts (Bravo-Rivera, Roman-Ortiz, Brignoni-Perez, Sotres-Bayon, & Quirk, 2014; Moscarello & LeDoux, 2013). A study has shown that the PrL region of the mPFC contributes to active avoidance in situations where conflict needs to be mitigated, but also contributes to the retention of conflict responses for reward (Capuzzo & Floresco, 2020). This may reveal that the suppression of infanticide by PVN to mPFC OT projections is a behavioral consequence of active conflict avoidance. In a study on pain in rats, OT neurons projections from the PVN to the PrL were found to increase the responsiveness of cell populations in the PrL, suggesting that OT may act by altering the local excitation-inhibition (E/I) balance in the PrL (Liu et al., 2023). A study on anxiety-related behaviors in male rats suggests that the anxiolytic effects of OT in the mPFC are PrL-specific but not infralimbic or anterior cingulate and that this is achieved primarily through the engagement of GABAergic neurons, which ultimately modulate downstream anxiety-related brain regions, including the amygdala (Sabihi, Dong, Maurer, Post, & Leuner, 2017). This finding may provide possible downstream pathways for further research.

(3) How groups of pup-care and infanticidal animals were created since there was no obvious pretest mentioned so perhaps there was the testing of a large number of animals until getting enough subjects in each group.

Before the experiments, we exposed the animals to pups, and subjects may exhibit pup care, infanticide, or neglect; we grouped subjects according to their behavioral responses to pups, and individuals who neglected pups were excluded.

(4) The apparent use of a 20-minute baseline data collection period for photometry that started right after the animals were stressed from handling and placement in the novel testing chamber.

In fiber photometric experiments, all experimental animals were required to acclimatize to the environment for at least 20 minutes prior to the experiment as described in the Methods section. The time 0 in Fig. 4 represents the point in time when a behavior or a segment of behavior started and is not the actual time 0 at which the test was started.

(5) A weakness in the results reporting is that it's unclear what statistics are reported (2 x 2 ANOVA main effect of interaction results, t-test results) and that the degrees of freedom expected for the 2 X 2 ANOVAs in some cases don't appear to match the numbers of subjects shown in the graphs; including sample sizes in each group would be helpful because the graph panels are very small and data points overlap.

Thanks for your suggestion. We displayed analysis methods for the data statistics and the sample sizes for each group of experiments in the figure legends.

The additional context that could help readers of this study is that the authors overlook some important mPFC and pup caregiving and infanticide studies in the introduction which would help put this work in better context in terms of what is known about the mPFC and these behaviors. These previous studies include Febo et al., 2010; Febo 2012; Peirera and Morrell, 2011 and 2020; and a very relevant study by Alsina-Llanes and Olazábal, 2021 on mPFC lesions and infanticide in virgin male and female mice. The introduction states that nothing is known about the mPFC and infanticide. In the introduction and discussion, stating the species and sex of the animals tested in all the previous studies mentioned would be useful. The authors also discuss PVN OT cell stimulation findings seen in other rodents, so the work seems less conceptually novel. Overall, the findings add to the knowledge about OT regulation of pup-directed behavior in male and female rodents, especially the PVN-mPFC OT projection.

We appreciate you very much to provide so many valuable references. We have cited them in the introduction and discussion. We agree with the reviewer’s opinion that nothing is known about the mPFC and infanticide is incorrect. It should be whether mPFC OT projections are involved in paternal cares and infanticide remains unclear. A study in mother rats indicated that inactivation or inhibition of neuronal activity in the mPFC largely reduced pup retrieval and grouping (Febo, Felix-Ortiz, & Johnson, 2010). In a subsequent study on firing patterns in the mPFC of mother rats suggested that sensory-motor processing occurs in the mPFC that may affect decision making of maternal care to their pups (Febo, 2012). In a study on new mother rats examining different regions of the mPFC (anterior cingulate (Cg1), PrL, IL), they identified a involvement of the IL cortex in biased preference decision-making in favour of the offspring (Pereira & Morrell, 2020). A study on maternal motivation in rats suggests that in the early postpartum period, the IL and Cg1 subregion in mPFC, are the motivating circuits for pup-specific biases (Pereira & Morrell, 2011), while the PrL subregion, are recruited and contribute to the expression of maternal behaviors in the late postpartum period (Pereira & Morrell, 2011).

**Reviewer #3 (Public Review):**
Summary:Here Li et al. examine pup-directed behavior in virgin Mandarin voles. Some males and females tend towards infanticide, others tend towards pup care. c-Fos staining showed more oxytocin cells activated in the paraventricular nucleus (PVN) of the hypothalamus in animals expressing pup care behaviors than in infanticidal animals. Optogenetic stimulation of PVN oxytocin neurons (with an oxytocin-specific virus to express the opsin transgene) increased pup-care, or in infanticidal voles increased latency towards approach and attack.Suppressing the activity of PVN oxytocin neurons promoted infanticide. The use of a recent oxytocin GRAB sensor (OT1.0) showed changes in medial prefrontal cortex (mPFC) signals as measured with photometry in both sexes. Activating mPFC oxytocin projections increased latency to approach and attack in infanticidal females and males (similar to the effects of peripheral oxytocin injections), whereas in pup-caring animals only males showed a decrease in approach. Inhibiting these projections increased infanticidal behaviors in both females and males and had no effect on pup caretaking.Strengths:Adopting these methods for Mandarin voles is an impressive accomplishment, especially the valuable data provided by the oxytocin GRAB sensor. This is a major achievement and helps promote systems neuroscience in voles.Weaknesses:The study would be strengthened by an initial figure summarizing the behavioral phenotypes of voles expressing pup care vs infanticide: the percentages and behavioral scores of individual male and female nulliparous animals for the behaviors examined here. Do the authors have data about the housing or life history/experiences of these animals? How bimodal and robust are these behavioral tendencies in the population?

As our response to reviewer 2, animals generally exhibit three types of behavioral responses toward pups, and data on the percentage of these different behavioral types occurring in the group will be included in another study in our lab. The reviewer's suggestion of scoring the behaviors is an inspiring idea that will help us to more fully parse these behaviors. Mandarin voles were captured from the wild in Henan, China. The experimental subjects were F2 generation voles reared in the Experimental Animal Centre of Shaanxi Normal University. In our observations, pup care and infanticide behaviors were conserved across several pup exposures, especially pup care behaviors, whereas for infanticide behaviors we did not conduct more pup exposures in order to protect the pups.

Optogenetics with the oxytocin promoter virus is a nice advance here. More details about their preparation and methods should be in the main text, and not simply relegated to the methods section. For optogenetic stimulation in Figure 2, how were the stimulation parameters chosen? There is a worry that oxytocin neurons can co-release other factors- are the authors sure that oxytocin is being released by optogenetic stimulation as opposed to other transmitters or peptides, and acting through the oxytocin receptor (as opposed to a vasopressin receptor)?

As reviewer suggested, more detailed information about virus construction and choice of optogenetic stimulation parameter have been added in the revised manuscript. The details about the construction of CHR2 and mCherry viruses used in optogenetic manipulation can refer to a previous study in which they constructed an rAAV-expressing Venus from a 2.6 kb region upstream of OT exon 1, which is conserved in mammalian species (Knobloch et al., 2012). For details about construction of the eNpHR 3.0 virus, expression of the vector is driven by the mouse OXT promoter, a 1kb promoter upstream of exon 1 of the OXT gene, which has been shown to induce cell type-specific expression in OXT cells (Peñagarikano et al., 2015). Details about the construction of OT1.0 sensor can be referred to the research of Professor Li's group (Qian et al., 2023). The mapping of the viral vectors and OT1.0 sensor is shown below.

The optogenetic stimulation parameters were used based on a previous study (He et al., 2021). However, our description of the parameters in the experiment is still not in detail, so some information about optogenetic stimulation parameters has been added in the method. In pupdirected pup care behavioral test, light stimulation lasted for 11 min. Parameters used in optogenetic manipulation of PVN OT neurons were ~ 3 mW, 20 Hz, 20 ms, 8 s ON and 2 s OFF and parameters used in optogenetic manipulation of PVN OT neurons projecting to mPFC were ~ 10 mW, 20 Hz, 20 ms, 8 s ON and 2 s OFF to cover the entire interaction. We performed fiber photometric experiments to determine the role that OT plays in behavior, and these results were able to support each other with optogenetic experiments. In addition, we further confirmed the role of optogenetic manipulation on OT release in combination with optogenetic inhibition and OT1.0 sensors (Supplementary Data Fig. 2). It has been previously shown that OT is able to act specifically on OTR in mPFC-PL (Sabihi et al., 2017). Our study focuses on oxytocin neurons as well as oxytocin release, and more research is needed to construct a more complex and complete network regarding the involvement of the OTR and other factors in the mPFC in these behaviors.

**Author response image 1. sa4fig1:** 

**Author response image 2. sa4fig2:** 

Given that they are studying changes in latency to approach/attack, having some controls for motion when oxytocin neurons are activated or suppressed might be nice. Oxytocin is reported to be an anxiolytic and a sedative at high levels.

As our response to reviewer 1, to exclude effects of locomotor activity on pup directed behaviors, we also investigated effect of optogenetic manipulations on the locomotor activity of experimental animals and found that optogenetic manipulation did not change levels of locomotor activity (Supplementary Data Fig. 6).

The OT1.0 sensor is also amazing, these data are quite remarkable. However, photometry is known to be susceptive to motion artifacts and I didn't see much in the methods about controls or correction for this. It's also surprising to see such dramatic, sudden, and large-scale suppression of oxytocin signaling in the mPFC in the infanticidal animals - does this mean there is a substantial tonic level of oxytocin release in the cortex under baseline conditions?

The optical fiber recording system used in the present study can automatically exclude effects of motion artifacts by simultaneously recording signals stimulated by a 405nm light source. As shown in the formula below, the z-score data were calculated and presented, and the increase and decline of the OT signal is a trend relative to the baseline. For a smooth baseline, the decreasing signal is generally amplified after calculation. In our experiments combining optogenetic inhibition and OT1.0 sensors, we were able to find that there was a certain level of OT release at baseline, on which there was room for a decrease in the signal recorded by the OT1.0 sensor.F0=Vbasal ¯ΔFF0=Vsignal −F0F0−Voff set σF=STD⁡(Vbasal)=Vsignal −F0σF

Figure 5 is difficult to parse as-is, and relates to an important consideration for this study: how extensive is the oxytocin neuron projection from PVN to mPFC?

AAV11-Ef1a-EGFP virus can infect fiber and retrogradely reach to cell body, thus this virus can be used to retrogradely trace neurons. We injected the this virus (green, AAV11-Ef1aEGFP) in the mPFC and observed virus infected and OT (red) positive neuron in the PVN (Yellow), and we also counted the OT neurons that project from PVN to mPFC and found that approximately 45.16% and 40.79% of cells projecting from PVN to the mPFC were OT-positive, and approximately 18.48% and 18.89% of OT cells in the PVN projected to the mPFC in females and males, respectively (Supplementary Data Fig. 4).

In Figures 6 and 7, the authors use the phrase 'projection terminals'; however, to my knowledge, there have not been terminals (i.e., presynaptic formations opposed to a target postsynaptic site) observed in oxytocin neuron projections into target central regions.

According your suggestion, we replaced the ‘terminals’ with ‘fibers’ to describe it more accurately..

Projection-based inhibition as in Figure 7 remains a controversial issue, as it is unclear if the opsin activation can be fast enough to reduce the fast axonal/terminal action potential. Do the authors have confirmation that this works, perhaps with the oxytocin GRAB OT sensor?

Thanks for your suggestion. We measured the OT release using OT1.0 sensors when the OT neuron projections in the mPFC were optogenetically inhibited. The result showed that optogenetic inhibition of OT neuron fibers in the mPFC significantly reduced OT release that validate the method of projection-based inhibition (Supplementary Data Fig. 5).

As females and males had similar GRAB OT1.0 responses in mPFC, why would the behavioral effects of increasing activity be different between the sexes?

In the present study, females released higher levels of OT into the mPFC (Figure 4 d, e) than males upon occurrence of different behaviors. In addition, females already exhibited more rapid approach and retrieval of pups than male before the optogenetic activation this may be the reason no effects of this manipulation were found in female.

**Recommendations for the authors:**

**Reviewer #1 (Recommendations For The Authors):**
(1) Check for spelling and grammar errors throughout.

Thanks to the reviewer's suggestion, we have checked and revised the article.

(2) Report effect sizes for all significant findings to allow evaluation of practical significance.

As reviewer suggested, we have added effect size both in the source data and in the main text, including d, η2 and odds ratio.

(3) Provide detailed information on the imaging parameters and analysis methods used in the Fos study.

The imaging parameters and analysis details for the Fos study have also been added in the revised manuscript. Brain slices of 40 µm thick were collected consecutively on 4 slides, each slide had 6 brain slices spaced 160 µm apart from each other. PVN area were determined based on the Allen Mouse Brain Atlas and our previous study, andFos, OT and merged positive neurons were counted.

(4) Compare the Fos colocalization results with previous studies examining PVN Fos and maternal/paternal care to contextualize the findings.

Our result about Fos and OT colocalization is consistent with previous study. In a previous study on virgin male prairie voles, OT and Fos colabeled neurons in the PVN increased after exposure to conspecific pups and experiencing paternal care (Kenkel et al., 2012). In another study of prairie voles, OT and c-fos colabeled neurons in PVN significantly increased after becoming parents which may be due to a shift from virgin to parents (Kelly et al., 2017).

(5) Discuss the limitations of the study, such as the potential effects of manipulating OT neurons on the release of other transmitters or the influence of other neurochemicals or brain regions on pupdirected behaviors, especially in females.

We agree with reviewer’s suggestion that several limitations should be discussed. Although we used a virus strategy to specifically activate or inhibit PVN OT neurons, other neurochemical may also be released during optogenetic manipulations because OT neurons may also release other neurochemicals. In one of our previous studies, activation of the OT neuron projections from the PVN to the VTA as well as to the Nac brain also altered pup-directed behaviors, which may also be accompanied by dopamine release (He et al., 2021). In addition, backpropagation of action potentials during optogenetic manipulations may also causes the same behavioral effect as direct stimulation of PVN OT cells. These effects on pup-directed behaviors should also be investigated further in the future study.

(6) Address the possibility of back-propagation of action potentials in the optogenetic manipulations causing the same behavioral effects as PVN OT cell stimulation.

We agree with the reviewer’s opinion hat optogenetic manipulation may possibly induce back-propagation of action potentials that may result in same behavioral effects as OT cell stimulation. We will pay attention to this issue in the future study.

(7) Investigate whether changes in locomotor behavior could explain the observed effects on pupdirected behaviors.

To exclude effects of locomotor activity on pup directed behaviors, we also investigated effect of optogenetic manipulations on the locomotor activity of experimental animals and found that optogenetic manipulation did not change levels of locomotor activity (Supplementary Data Fig. 6).

(8) Report the percentage of PVN->mPFC neurons labeled that were OT-positive.

AAV11-Ef1a-EGFP virus can infect fiber and retrogradely reach to cell body, thus this virus can be used to retrogradely trace neurons. We injected this virus (green, AAV11-Ef1a-EGFP) in the mPFC and observed virus infected and OT (red) positive neuron in the PVN (Yellow), and we also counted the OT neurons that project from PVN to mPFC and found that approximately 45.16% and 40.79% of cells projecting from PVN to the mPFC were OT-positive, and approximately 18.48% and 18.89% of OT cells in the PVN projected to the mPFC in females and males, respectively (Supplementary Data Fig. 4).

(9) Directly compare the sexes in the behavioral analysis and discuss any potential sex differences.

We agree with the reviewer's suggestion and have added comparisons between two sexes and discussion about relevant results.

(10) If available, report and discuss the OT expression levels and the number of OT neurons manipulated in each sex.

In the present study, we have counted the number of OT cells, but did not measure the level of OT expression using WB or qPCR. In addition, the percentages of CHR2(H134R) and eNpHR3.0 virus infected neurons in total OT positive neurons were presented (Supplementary Data Fig. 7), but we did not know how many cells were actually manipulated during the optogenetic experiment.

(11) Expand the discussion to include what could be regulating or interacting with the OT circuit under investigation, particularly in females where the effects were less pronounced.

As the reviewers suggested, we have also added relevant discussion. In addition to OT neurons, OTR neurons may also regulate behavioral responses to pups. In a study of virgin female mice pup exposure was found to activate oxytocin and oxytocin receptor expressing neurons (Okabe et al., 2017). Other brain regions such as preoptic area (POA) may also be involved in parental behaviors. For example, virgin female mice repeatedly exposed to pups showed shorter retrieval latencies and greater c-Fos expression in the preoptic area (POA), concentrations of OT in the POA were also significantly increased, and the facilitation of alloparental behavior by repeated exposure to pups occurred through the organization of the OT system (Okabe et al., 2017). A recent study suggests that OT of the PVN is involved in the care of pups by male voles (He et al., 2021). This study suggests that PVN to ventral tegumental area (VTA) OT projections as well as VTA to nucleus accumbens (NAc) DA projections are involved in the care of pups by male voles. Inhibition of OT projections from the PVN to the VTA reduces DA release in the NAc during licking and grooming of pups (He et al., 2021).

**Reviewer #2 (Recommendations For The Authors):**
A few additional things the authors may want to consider:(1) I don't understand the subject numbers in the peripheral OT study data shown in Figure 8. Panels p and q have 69 females shown and 50 males. Was there a second, much larger, IP injection study conducted that was different than the subjects shown in panels a-o that had ~5 subjects per treatment group per sex?

Sorry for the confusing. More animals were used to test effects of OT on infanticide behaviors in our pre-test. These data combined with data from formal pharmacological experiment were presented in Fig. 8p, q. After OT treatment, the changes in detailed and specific behaviors were only collected in several animals. We have clarified that in the revised manuscript.

(2) The authors suggest higher baseline OT release in the female mPFC, which makes sense and helps explain some of their results. It seems that the data in Figure 1 show what is probably no sex difference in OT cell numbers in the PVN of Mandarin voles, which is unlike the old studies in mice or rats. If readers look at the data in Figure 1 showing what seems to be no sex difference in OT cell number, the authors' argument in the discussion about mPFC OT release levels higher in females would be inconsistent with their own data shown. The authors have the brain sections they need to help support or undermine this argument in the discussion, so maybe it would be useful to analyze the OT cell numbers across the PVN and report it in this paper or briefly mention it in the discussion.

We compared the numbers of OT neurons, activated OT neurons (OT and Fos doublelabeled neurons) and level of OT release between males and females. We found that females have more activated OT neurons (Figure1, d, g) and released higher levels of OT into the mPFC (Figure 4 d, e) than males. This part has been added in the result and discussion. The inconsistency of the OT cell numbers with previous studies may be due to the method of cell counting, as we did not count all slides consecutively.

(3) The discussion suggests visual cues are involved in mPFC OT release relevant for pup care or infanticide, but this is a very odd claim for nocturnal animals that live and nest with their pups in underground burrows.

Sorry for the confusing. Here, we cited the finding in mice that activation of PVN OT neurons induced by visual stimulation promoted pup care to support our finding that the activity of OT cells of the PVN is involved in pup care, rather than to illustrate the role of visual stimulation in voles. We have clarified that in the revised manuscript.

(4) The lack of decrease in mPFC OT release in the 2nd and 3rd approaches to pups is probably because the release was so high after the 1st approach that it didn't have time to drop before the subsequent approaches. The authors don't state how long those between-approach intervals were on average to help readers interpret this result.

As described in our methods, we spaced about 60 s between each behavioral test to allow the signal return back to the baseline level.

(5) Do PVN-mPFC OT somata collateralize to other brain sites? Could mPFC terminal stimulation activate entire PVN cells and every site they project to? A caveat could be mentioned in the discussion if there's support for this from other optogenetic and PVN OT cell projection studies.

We verified the OT projections from PVN to mPFC, to validate the optogenetic manipulation of this pathway, but did not investigate whether the OT neurons projecting from PVN to mPFC also project collaterally to other brain regions. It is suggested that mPFC terminal stimulation only activate PVN OT cells projecting mPFC, whether other OT neurons were activated remains unclear.

(6) I don't see an ethics statement related to the experiments obviously having to involve pup injury or death. Nothing is said in methods about what happened after adult subjects attacked pups. I assumed the tests were quickly terminated and pups euthanized.

In case the pups were attacked, we removed them immediately to avoid unnecessary injuries, and injured pups were euthanized.

(7) The authors could be more specific about what psychological diseases they refer to in the abstract and elsewhere that are relevant to this study. Depression? Rare cases of psychosis? Even within the already rare parental psychosis, infanticide is tragic but rare.

Infanticide is caused by a variety of factors, mental illness, especially depression and psychosis, is often a very high risk factor among them (Milia & Noonan, 2022; Naviaux, Janne, & Gourdin, 2020). In human, infanticide has been used to refer to the killing, neglect or abuse of newborn babies and older children (Jackson, 2006). Here, we believe that research on the neural mechanisms of infanticide can also contribute to the understanding and treatment of attacks on children, physical and verbal abuse, and direct killing of babies.

(8) Figure 8 - in one case the "*" is a chi-square result , correct?

Thanks for your careful checking. In Figure 8p, q, we applied the chi-square test and added it in the legend.

**Reviewer #3 (Recommendations For The Authors):**
The only other thing is a typo on line 135: the authors mean 'stimulation' instead of 'simulation'.

Corrected.

References

Bravo-Rivera, C., Roman-Ortiz, C., Brignoni-Perez, E., Sotres-Bayon, F., & Quirk, G. J. (2014). Neural structures mediating expression and extinction of platform-mediated avoidance. J Neurosci, 34(29), 9736-9742. doi:10.1523/jneurosci.0191-14.2014

Capuzzo, G., & Floresco, S. B. (2020). Prelimbic and Infralimbic Prefrontal Regulation of Active and Inhibitory Avoidance and Reward-Seeking. J Neurosci, 40(24), 4773-4787. doi:10.1523/jneurosci.0414-20.2020

Febo, M. (2012). Firing patterns of maternal rat prelimbic neurons during spontaneous contact with pups. Brain Res Bull, 88(5), 534-542. doi:10.1016/j.brainresbull.2012.05.012

Febo, M., Felix-Ortiz, A. C., & Johnson, T. R. (2010). Inactivation or inhibition of neuronal activity in the medial prefrontal cortex largely reduces pup retrieval and grouping in maternal rats. Brain Res, 1325, 77-88. doi:10.1016/j.brainres.2010.02.027

He, Z., Young, L., Ma, X. M., Guo, Q., Wang, L., Yang, Y., . . . Tai, F. (2019). Increased anxiety and decreased sociability induced by paternal deprivation involve the PVN-PrL OTergic pathway. Elife, 8. doi:10.7554/eLife.44026

He, Z., Zhang, L., Hou, W., Zhang, X., Young, L. J., Li, L., . . . Tai, F. (2021). Paraventricular Nucleus Oxytocin Subsystems Promote Active Paternal Behaviors in Mandarin Voles. J Neurosci, 41(31), 66996713. doi:10.1523/jneurosci.2864-20.2021

Jackson, M. (2006). Infanticide. The Lancet, 367(9513), 809. doi:https://doi.org/10.1016/S01406736(06)68323-2

Kelly, A. M., Hiura, L. C., Saunders, A. G., & Ophir, A. G. (2017). Oxytocin Neurons Exhibit Extensive Functional Plasticity Due To Offspring Age in Mothers and Fathers. Integr Comp Biol, 57(3), 603618. doi:10.1093/icb/icx036

Kenkel, W. M., Paredes, J., Yee, J. R., Pournajafi-Nazarloo, H., Bales, K. L., & Carter, C. S. (2012). Neuroendocrine and behavioural responses to exposure to an infant in male prairie voles. J Neuroendocrinol, 24(6), 874-886. doi:10.1111/j.1365-2826.2012.02301.x

Knobloch, H. S., Charlet, A., Hoffmann, L. C., Eliava, M., Khrulev, S., Cetin, A. H., . . . Grinevich, V. (2012). Evoked axonal oxytocin release in the central amygdala attenuates fear response. Neuron, 73(3), 553-566. doi:10.1016/j.neuron.2011.11.030

Liu, Y., Li, A., Bair-Marshall, C., Xu, H., Jee, H. J., Zhu, E., . . . Wang, J. (2023). Oxytocin promotes prefrontal population activity via the PVN-PFC pathway to regulate pain. Neuron, 111(11), 17951811.e1797. doi:10.1016/j.neuron.2023.03.014

Mei, L., Yan, R., Yin, L., Sullivan, R. M., & Lin, D. (2023). Antagonistic circuits mediating infanticide and maternal care in female mice. Nature, 618(7967), 1006-1016. doi:10.1038/s41586-023-061479

Milia, G., & Noonan, M. (2022). Experiences and perspectives of women who have committed neonaticide, infanticide and filicide: A systematic review and qualitative evidence synthesis. J Psychiatr Ment Health Nurs, 29(6), 813-828. doi:10.1111/jpm.12828

Moscarello, J. M., & LeDoux, J. E. (2013). Active avoidance learning requires prefrontal suppression of amygdala-mediated defensive reactions. J Neurosci, 33(9), 3815-3823. doi:10.1523/jneurosci.2596-12.2013

Murugan, M., Jang, H. J., Park, M., Miller, E. M., Cox, J., Taliaferro, J. P., . . . Witten, I. B. (2017). Combined Social and Spatial Coding in a Descending Projection from the Prefrontal Cortex. Cell, 171(7), 1663-1677.e1616. doi:10.1016/j.cell.2017.11.002

Naviaux, A. F., Janne, P., & Gourdin, M. (2020). Psychiatric Considerations on Infanticide: Throwing the Baby out with the Bathwater. Psychiatr Danub, 32(Suppl 1), 24-28.

Okabe, S., Tsuneoka, Y., Takahashi, A., Ooyama, R., Watarai, A., Maeda, S., . . . Kikusui, T. (2017). Pup exposure facilitates retrieving behavior via the oxytocin neural system in female mice. Psychoneuroendocrinology, 79, 20-30. doi:10.1016/j.psyneuen.2017.01.036

Peñagarikano, O., Lázaro, M. T., Lu, X. H., Gordon, A., Dong, H., Lam, H. A., . . . Geschwind, D. H. (2015). Exogenous and evoked oxytocin restores social behavior in the Cntnap2 mouse model of autism. Sci Transl Med, 7(271), 271ra278. doi:10.1126/scitranslmed.3010257

Pereira, M., & Morrell, J. I. (2011). Functional mapping of the neural circuitry of rat maternal motivation: effects of site-specific transient neural inactivation. J Neuroendocrinol, 23(11), 1020-1035. doi:10.1111/j.1365-2826.2011.02200.x

Pereira, M., & Morrell, J. I. (2020). Infralimbic Cortex Biases Preference Decision Making for Offspring over Competing Cocaine-Associated Stimuli in New Mother Rats. eNeuro, 7(4). doi:10.1523/eneuro.0460-19.2020

Qian, T., Wang, H., Wang, P., Geng, L., Mei, L., Osakada, T., . . . Li, Y. (2023). A genetically encoded sensor measures temporal oxytocin release from different neuronal compartments. Nat Biotechnol, 41(7), 944-957. doi:10.1038/s41587-022-01561-2

Sabihi, S., Dong, S. M., Maurer, S. D., Post, C., & Leuner, B. (2017). Oxytocin in the medial prefrontal cortex attenuates anxiety: Anatomical and receptor specificity and mechanism of action. Neuropharmacology, 125, 1-12. doi:10.1016/j.neuropharm.2017.06.024